# Instability Challenges and Stabilization Strategies of Pharmaceutical Proteins

**DOI:** 10.3390/pharmaceutics14112533

**Published:** 2022-11-20

**Authors:** Mohsen Akbarian, Shu-Hui Chen

**Affiliations:** Department of Chemistry, National Cheng Kung University, Tainan 701, Taiwan

**Keywords:** protein drugs, antibody drugs, antibody drug conjugates, pharmaceutical proteins and peptides, stabilization, denaturing stresses, protein aggregation, protein folding, protein drug characterizations, LC-MS

## Abstract

Maintaining the structure of protein and peptide drugs has become one of the most important goals of scientists in recent decades. Cold and thermal denaturation conditions, lyophilization and freeze drying, different pH conditions, concentrations, ionic strength, environmental agitation, the interaction between the surface of liquid and air as well as liquid and solid, and even the architectural structure of storage containers are among the factors that affect the stability of these therapeutic biomacromolecules. The use of genetic engineering, side-directed mutagenesis, fusion strategies, solvent engineering, the addition of various preservatives, surfactants, and additives are some of the solutions to overcome these problems. This article will discuss the types of stress that lead to instabilities of different proteins used in pharmaceutics including regulatory proteins, antibodies, and antibody-drug conjugates, and then all the methods for fighting these stresses will be reviewed. New and existing analytical methods that are used to detect the instabilities, mainly changes in their primary and higher order structures, are briefly summarized.

## 1. Introduction

It has been a century since the first peptide drug (insulin) entered the health sector [1]. Currently, there are a variety of proteins and peptides such as growth hormone, green fluorescence protein, insulin analog forms, and many antibodies in the field of therapy, diagnosis, control and management of diseases, some of which are intentionally very different from their native/wild-type counterparts [2]. The protein drug market is divided into several segments: antibody drugs, peptide hormones, blood products, and enzymes [3]. The antibody drugs sector has experienced some of the fastest growth in this marketplace. In addition to being used as therapeutic proteins against various diseases, antibody molecules have also been conjugated with various anticancer drugs, named antibody-drug conjugates (ADCs), to increase drug efficacy by specific targeting. Moreover, the advances made in drug delivery systems using nanotechnology and the rapid increase in chronic diseases are among the growing causes of this market [4]. However, not everything is positive and clear about these protein-based drugs; there are other challenges such as the synthesis or production methods of these drugs [5], purification from expression systems [6], correct folding into native forms in the process of production and consumption [7], and instabilities of these drugs during their manufacturing, storage, and delivery. Among these, structural instability caused by unfolding, misfolding, or covalent/non-covalent modifications, as well as aggregation can degrade drug efficacy and greatly overshadow the bright features of these protein-based drugs.

Protein misfolding or incorrect folding is an unwanted instability process in which the protein either gets function (triggering gain-of-function diseases) and/or loses its specific function (initiating loss-of-function diseases) [8]. Likewise, the in vitro misfolding of recombinant proteins/peptides is also another challenge in the field of biotechnology and the production of pharmaceutical proteins. Figure 1 indicates the different fates of a protein after its production by a ribosome [9]. From the causative point of view, as mentioned above, several factors trigger misfolding of a protein [10]. 

At the beginning of the transcription of a protein, thanks to the intracellular chaperone systems and the biophysical laws governing protein folding, correct folding occurs most of the time. When due to cellular defects and rapid protein expression, protein folding becomes problematic, and several fates may occur for the protein. In cases where misfolding leads to the loss of protein activity (such as enzymes), the corresponding disease will appear directly. As this state continues, the misfolded protein may turn into amorphous clots or aggregates with regular structures, each of which can lead to various neurological diseases or even cancer. In the most optimistic scenario, the misfolded protein enters the proteasome machinery and is initially converted into smaller peptides and finally broken down into building amino acids. In a pessimistic view, in some cases, diseases may occur from the peptides obtained from the proteasome system, which is very rare.

On the other hand, proteins have a great tendency to aggregate under various stresses, especially when they are kept at high concentrations required for effective doses. Therefore, the main problem in the production of pharmaceutical proteins, is the issue of stability with the hope of maintaining their activity [8]. Types of strategies that have been adopted to increase the stability of these biomacromolecules include: direct mutations on the structure of the genome-encoding proteins [9], joining these molecules to inert proteins (such as human serum albumin) [10], their chemical changes after conjugation (like ADCs) or purification from expression systems [11], the use of nanoparticles, polymers, sugars and other additives to maintain the structure of proteins and drug peptides [12], and the engineering of storage containers and packing approaches [13] for these large molecules. Analytical methods to detect and characterize highly heterogeneous structural changes of protein-based drugs are other challenging issues in tackling the instability of these drugs [14]. 

This review will discuss various stresses causing instabilities and available strategies for improving the stability of pharmaceutical proteins and peptides. Analytical methods used to monitor and characterize pharmaceutical proteins and drug peptides will also be described. Here, pharmaceutical proteins include protein therapeutics and proteins which are not the main therapeutics ingredients but aim to improve the effectiveness of drug targeting (like ADCs) or stabilization in blood circulation (like serum albumin). 

## 2. Stability Issues of Different Protein-Based Drugs

Of the significant aspects in the industrialization of proteins are their stability and solubility [14,15]. The stability of proteins during the process of expression and purification is one of the most important and challenging issues because many recombinant proteins are unstable under the conditions they are expressed and lose their correct folding or undergo proteolytic digestion [16]. Therefore, studying the stability of proteins and identifying the factors that increase their stability is one of the important and interesting topics in scientific societies [17]. Today, numerous plans have been proposed to increase the solubility and stability of proteins, which include guided changes in the protein molecule or optimization in the expression protocols [18], purification and solubility process of proteins [19]. 

The production of biological medicine is not a direct path, because, on the one hand, this process requires a lot of work, and on the other hand, the issue of product instability during the process and storage is raised. In addition, there are other challenges such as the need for strong laws, scrutinizing Good Manufacturing Practice (GMP) regulations and intense competition in the consumer market (for example due to the emergence of biosimilars) [20]. Proteins and peptides are only stable in a limited range of concentration, temperature, ionic strength and acidity conditions (marginal stability) and are very prone to physical and chemical destabilizations. Many recombinant proteins are also under conditions that become unstable and lose their correct folding. Additionally, medicinal proteins have a great tendency to aggregate, especially when they are in high concentration. When talking about the stability of protein-based drugs, researchers often seek to understand why and when the native structure of proteins tends to become denatured. For more than 50 years, scientists have made great efforts to understand the relationship between protein denaturation and the competition between their reversible and irreversible denaturations [21]. The concern of stability is different for different classes of proteins. Some protein drugs have an enzymatic activity that can replace abnormal and low-expressed counterparts in the body. Since enzymes usually have a high turnover number, a small disorder in their structure will lead to huge changes in the accumulation of their substrate and, on the other hand, the concentration of their products [22]. Consequently, for this category of proteins, not only enzyme aggregation but also the smallest change in the structure can have great effects [23]. 

Monoclonal antibodies (mAbs) accounted for almost half (48%) of the therapeutics protein sales in recent years [24]. Since all the antibodies have a binding role to the antigen, the most important components in this category are maintaining the structure of complementarity-determining regions (CDRs) which bind their specific antigens. Most of the stability challenges in this field are related to their aggregation and oxidation in different conditions and high on-demanded concentrations [25]. In addition, ADCs [26] are a special class of protein-based drugs made by binding chemical drugs (payloads) to antibodies with an emphasis on payload efficacy. The concept is to achieve high cancer cell specificity and low cytotoxicity to off-target cells through antibody targeting so that a broader therapeutic window of the payload can be gained [27]. This idea was introduced in early 1970. By now, 12 ADCs have been approved by the FDA [28,29] and about 60 other ADCs are also in clinical trials [30]. One of the critical efforts that need to be harmonized in this regard is the issue of stabilization of these important therapies. This is because the binding of a chemical drug may give the antibody a new behavior that overshadows its stability and leads to the changing of native antibodies to unfolded states [29]. 

## 3. Causes of Instability

### 3.1. Physical Instability

#### 3.1.1. Temperature-Induced Instability

Some proteins are very resistant to high temperatures [31], nonetheless, most of them are unable to maintain their natural structure against this physical stress and turn into aggregated states [32]. In this process, it is suggested that the hydrophobic domains of the proteins, which are mainly hidden in the protein structure (by van der Waals and hydrogen forces), are initially removed from the buried state and exposed to the external environment (surface of protein). Due to the hydrophobic nature of these domains, hydrophobic interactions occur and lead to the binding of a large number of protein molecules, which ultimately leads to the formation of aggregated states of the protein. For ADCs, the hydrophobicity of the payload as well as the hydrophobicity of the antibody surface will be one of the most important factors in shortening the shelf life of the final product of ADCs [33]. Nevertheless, unlike antibodies, the nature of payloads tends to be more hydrophobic, which has led to a challenge to connect these two different natures (connecting the hydrophobic payload to the hydrophilic surface of the antibody) [34]. As a result, maintaining hydrophobic structures while inactivated by temperature is very important to suppress or modulate the consequences of the phenomena [35,36]. 

Figure 2 shows the relationship between temperature and protein unfolding. The maximum of the ΔG_unfolding_ is in a small temperature range, after which a protein instability occurs beyond this certain temperature (either below or above this temperature) [37]. The width of the diagram or the sharpness of the peak determines how well the protein can withstand temperature changes. The wider the graph, the higher the resistance of the protein to temperature changes [38]. Not only increasing the temperature of a protein has the potential to unfold it, but lowering it to a critical point can be a lament of protein aggregation [39]. The ribonuclease A enzyme has been shown to precipitate at temperatures as low as −22 °C [40], and temperatures below zero have been reported to be destructive to serum albumin [41]. Nevertheless, thanks to poor hydrophobic interactions, protein unfolding is less likely to occur at low temperatures. This is probably why cold aggregation is usually reversible. It has been observed that the aggregation of serum cryoglobulins protein at a temperature below 37 °C can be reversibly converted to a soluble state and it has also been reported that the human IgM cryoglobulin in the temperature window between 10–12 °C turns into a jelly from which with increasing temperature this state becomes reversible [38]. The discussion of protein aggregation at low temperatures will be interpreted in another section of this article.

Whether protein aggregates in the aggregation-driven force (Figure 2B) or can survive as a refolded state is most likely related to the collapsing of hydrophobic domains and the repulsion of electrostatic forces. Thus, at high temperatures, the collapse of hydrophobic forces is overcome in competition with the electrostatic repulsion and finally, the protein will aggregate [32]. Apart from the hydrophobic challenge, it has also been reported that increasing the temperature of the protein solution leads to an increase in the rate of protein oxidation and deamidation, which in itself leads to faster protein aggregation [42].

A quantity index also defines the temperature stability of proteins, which is called the melting temperature (Tm) of proteins. This temperature is equivalent to the temperature at which half of the protein molecules are unfolded. Unsurprisingly, this temperature varies between different proteins, and in nature, it usually fluctuates between 40 and 80 °C [43]. In many stages of production, purification, packaging and transporting of medicinal proteins, it is essential to keep the temperature of the products below the agreed melting point. By reaching the maximum energy state (also the most unstable state of the protein), with a faster rate of activation, the target protein falls into the valley of energy and goes to the point of unfolding. Between these pathways, many intermediate states may happen for the protein, each of which can be individually stable or unstable. Subsequently, as the unfolding process continues, communities of unfolded proteins accumulate regularly or irregularly, eventually turning into morphous and/or amorphous aggregated states [44] (Figure 2B). 

#### 3.1.2. Cold Denaturation

The history of studying low temperatures on the structure of proteins probably dates back to the 1930s and Hopkins’s article [45]. In the study, it was found that in the presence of urea, the ovalbumin protein unfolds at 0 °C faster than at 23 °C. Clark later, proved this consequence in his observations, but stated that this result can be seen only in the presence of high concentrations of urea while in low concentrations of urea, the unfolding rate increases with increasing temperature [46]. 

It has been described that cold-induced unfolding is mainly due to a change in the pattern of interaction between water molecules and non-polar groups in the protein [47]. As the temperature decreases, the free energy of the interaction decreases, which increases the conditions for this unwanted communication and ultimately leads to the hydration of the hydrophobic domains [48]. With the help of in silico studies, it has been proven that with decreasing temperature, the diameter of this aqueous layer increases around non-polar (hydrophobic) groups, and fortunately, with increasing temperature, this harmful aqueous layer disappears, which is a reason for the reversibility of cold denaturation of most proteins [49].

The vast majority of studies have referred to the law of all-or-none during low temperature-induced unfolding. This means that during the process, there are two important states of protein: unfolded and folded state [50]. Nevertheless, low-temperature NMR studies by Wand et al., indicated that this process is step-by-step and produces intermediates of semi-active structures in addition to completely inactive configurations [51]. Aside from all these cold-denaturation issues, fortunately for most proteins, cold-temperatures are dangerous below freezing points [51].

Some early studies in the field of cold unfolding have pointed out the importance of the effect of pressure on this phenomenon [52,53]. As seen at high pressures, the cold unfolding process accelerates. Meanwhile, it has been reported that when the pressure of the bovine pancreatic ribonuclease A solution rises above the pressure where water transforms to the dense ice phase (2 kbars), the local density of water around the polar groups of the protein decreases (by reducing the temperature), leading to the unfolding of the protein [53]. In this section, with the help of Figure 3 diagram, the importance of the relationship between temperature and pressure during protein unfolding is discussed.

Most proteins remain in their natural state (folded) within a certain range of temperature and pressure. For most of them, the critical temperature is 50 °C. In addition, in negative temperatures, the change of the normal state to unfolding has been observed, which, unlike heat-induced unfolding, cold-induced unfolding is reversible. Moving toward this temperature window, protein unfolding will occur with little stress. At room temperature, the highest pressure is required for the protein to transition from the fold to unfold.

#### 3.1.3. Photo-Induced Instability

All aspects of life are constantly and/or temporarily exposed to light, and on the occasion of light-induced problems, living organisms within the cell have a mechanism in place to repel these damages. Although the protective layer of the atmosphere absorbs most of the dangerous UV waves (UVC: less than 295 nm), in the habitable atmosphere around us, UVB and A waves, as well as visible light, is also dangerous for bio-infrastructures. It is generally suggested that light-induced damage in proteins occurs by two general mechanisms; direct and indirect pathways [55,56,57]. The photo-induced damages may also be regarded as chemical instability since chemical conversions are involved in these pathways.

The direct pathway, which occurs mainly with UVB waves (λ 280–320 nm), is restricted to a small number of amino acids in proteins such as tyrosine, tryptophan, cysteine, and histidine. Once the UVB energy is received by these chromophores (the residues), they temporarily change to the first-excited singlet states. This state is transient and quickly turns into a triplet state which is relatively more stable with the release of excess energy. The released energy can be transferred to other functional groups. The resulting intermediates in the triplet state may generate dangerous free electrons that can interact with other functional groups and, as a radical group, disrupt protein function [57,58]. One of the problems is the release of hydrated electrons, which are regarded as one of the reactive oxidizing species (ROS) and are well described in Refs. [59,60]. These hydrated electrons can attack the free carbonyls at the amine end or the side chain of the amino acid lysine, leading to deamination and/or the elimination of H˙. It has also been described that the production of anionic radicals in small peptides originates from these hydrated electrons [61]. On the other hand, the indirect path of photo-induced damage is due to the formation of singlet oxygen, which occurs due to the energy transfer to intra-protein bonds and/or external chromophores. The latter include porphyrins [62], drugs, polyaromatic, and dye-conjugated molecules [57,63]. All of the above can lead to structural changes in the protein and, ultimately, loss of function. 

To summarize what has been believed about the effect of photos on proteins/peptides, the following schematic (Figure 4) is presented.

Some intrinsic factors of proteins such as their amino acid sequence and even their spatial structure can affect vulnerability to photo stress. In addition to the intrinsic properties, environmental factors can also affect the vulnerability of the protein to stress.

In the line with the above information, the effects of photo on ADCs can be much more dangerous, especially through the indirect path of photo-induced damage to the monoclonal structure of antibodies since some conjugated small molecules are themselves chromophores which can interact with UV to make ADCs more susceptible to unfolding compared with non-conjugated antibody counterparts. For example, in the case of trastuzumab, the intact protein itself has long-term physical and chemical stabilities. Conversely, when combined with eosin, as a model for chemical drugs, it becomes more sensitive to photo-induced aggregation (the aggregation was started after 7 min incubation of eosin conjugated trastuzumab under 20 W·h/m^2^ of UVA light) [56]. Doxorubicin (a topoisomerase II inhibitor), which is used in a wide range of ADCs [64,65], is also very sensitive to light; once exposed to UVA waves, it either breaks down or leads to the production of ROS [66]. The latter byproduct is capable of threatening the protein portion of ADCs for aggregation. There are other drugs similar to Dox used for ADCs, although they have also been shown to be highly sensitive to light. Among these drugs, cyclopropapyrroloind-4-one, CC-1065, and cyclopropabenzindol-4-one are the most important candidates in which light’s destructive role on protein structure should be seriously considered [67].

#### 3.1.4. Agitation-Induced Instability 

Agitation during stages of pumping, mixing, shipping, filtering, and filling can affect the structure of the therapeutic proteins and lead to aggregation [68,69].

The role of turbidity/agitation of the aquatic environment around the protein in the kinetics of protein aggregation is very complex and debatable. Many studies mention that different types of turbulence caused in different ways have diverse effects on the amount of aggregation. For example, it has been demonstrated that that the effect of steering on the speed and percentage of aggregation is more pronounced than shaking [70], and even the location of the steering can affect this quantity [71]. In other studies, researchers even went so far as to point out the role of dropping in the process of protein aggregation. A dropping method is sometimes used when filling therapeutic vials and syringes with protein drugs. By dropping the products, bubbles are formed, which are supposed to form oxygen free radicals near these bubbles, which lead to a change in the structure of the protein and eventually causing its aggregation [72,73].

Looking at the mechanism of the effect of agitation on protein aggregation, some researchers believe that due to the increase in the liquid and air interfaces during the enforced agitation, the protein aggregation rate increases [74,75]. This effect has been shown to increase the rate of aggregation and amyloid formation in proteins such as amyloid-β [76,77], insulin [21], and β-lactoglobulin [78] more than three times on average. For a visual look at how environmental agitation affects the unfolding of proteins, Figure 5 is drawn.

With the increase of the environmental agitation, the interface surface between the liquid and the air as well as the liquid with the body’s container (solid) will increase. Air, as a hydrophobic environment, leads to the exposure of the hydrophobic parts of a protein from inside the structure, and the same happens for the interaction between proteins on the solid surface. With the continuation of this process, most of the folded proteins become unfolded, which leads to the reduction of the active protein population, then aggregation due to environmental agitation.

Using hydrophobic excipients, in several independent studies, there are other theories for how proteins are aggregated during agitation stress. For example, in the study of Bam et al., by the use of tween, it was found that hydrophobic interactions can modulate the aggregation rate of human growth hormone, indicating hydrophobic connections between unfolded proteins in agitation stress [81]. Similarly, others have also used additives such as salts [82,83], lipids [84], surfactants [85,86], and various additives [87] to reduce the aggregation of proteins in the hope of reducing the interface of air and solution created during agitation.

### 3.2. Chemical Instability

#### 3.2.1. Hydrolysis 

Since almost all parts of life form and continue to survive in the aquatic environment, the phenomenon of occasional hydrolysis is ubiquitous. In many biological processes, this phenomenon, which is defined as breaking the chemical bond by water molecules, is common and useful, but in some cases, such as the storage of drug proteins in water, it can be dangerous and lead to some unpleasant consequences such as severe allergies [88]. Many proteins, including therapeutic monoclonal antibodies, have been reported to be non-enzymatically hydrolyzed in the hinge region [89,90]. Cordoba et al. found that incubating IgG1 antibody in a neutral, aqueous medium for three months resulted in some non-enzymatic hydrolysis [91]. With the efforts of Dillon et al. using a series of reverse phase chromatography linked with mass spectroscopy (LC-MS), the same result was obtained for the IgG1 antibody [92]. As a result, in both acidic and alkaline conditions, the rate of this reaction increases [93], which is mainly due to the protonation of carbonyl oxygen or the addition of a hydroxyl group to the carbonyl carbon in the peptide bond. However, by studying several dipeptides in neutral conditions, it has been indicated that it is necessary to overcome a large energy barrier equal to 27–30 kcal/mol for hydrolysis to occur [94]. Nutritionally speaking, hydrolysis of a protein can also lead to generating and exposing epitopes in the structure of the protein, which can eventually cause various allergies. It has been revealed that different types of milk proteins with different hydrolysis percentages can bring about percentages of allergenic shocks in consumers [95]. 

Concerns about the consequences of hydrolysis of protein drugs are not limited to the proteins themselves. Rather, additives are sometimes used during the formulation of these drugs; hydrolysis can lead to the deformation of the protein. As a clear example, when the monoclonal antibody MAB001 was examined in terms of susceptibility to aggregation with different sugars, it was found that this response is higher in the presence of sucrose than sorbitol, which also depends on the temperature and pH of the environment. Further studies have shown that this destructive effect is due to the hydrolysis of sucrose and the effect of the byproducts of hydrolyzed sucrose on the protein [96]. Therefore, in addition to the structural properties of the protein, the formulation of the therapeutic protein should also be considered, avoiding the destructive effects of hydrolysis.

#### 3.2.2. Oxidation

Today, many pharmaceutical proteins are produced by recombinant DNA technology using bacterial or mammalian cells like Chinese hamster ovary (CHO) expression systems [97]. The oxidation and eventual breakdown of proteins may even occur in the expression platform [97]. For example, the G1 antibody in the hamster expression system has been shown to degrade at rapid oxygen delivery rates, producing a pattern similar to the incubation of antibodies to hydrogen peroxide in an in vitro condition [98]. In addition to the challenges of production, the purification process of pharmaceutical proteins sometimes leads to oxidation. For example, during the purification of lactate dehydrogenase using metal affinity chromatography, degradation was observed due to the protein oxidation induced by metal ions [99].

Besides, many other factors can lead to protein oxidation and subsequent decomposition, including exposure to oxygen during shipping and storage, the presence of some oxidants, light oxidation, and the presence of transition metals in formulation steps [100]. Methionine is the most susceptible amino acid residue to oxidation [101,102]. In the case of monoclonal antibodies, there are two methionine residues in their FC (crystallizable fragment) region, which have caused the greatest weakness points of this group of protein drugs. The two residues are located around the neonatal Fc receptor (FcRn) binding region; any change in which could lead to a change in the behavior of the monoclonal receptor or a change in their cytotoxicity [103,104]. In addition to methionine, it has been observed that other amino acids such as tryptophan, histidine, cysteine, tyrosine, and sometimes amino acids arginine, lysine, and proline are also involved in the oxidation of proteins [105]. On top of problems with protein structure, oxidation of these amino acids sometimes results in immunological reactions [106,107]. Some studies have revealed that the oxidation process rapidly takes place between the skin and the blood before it enters the bloodstream, once subcutaneously injected so that dangerous by-products enter the bloodstream [108]. The extent of oxidation could be greatly affected by the conjugation sites of ADCs. The cysteine-conjugated payload showed the highest sensitivity to oxidation damage compared to the lysine-conjugated payload. It was also observed that due to the different structural changes of the cysteine-conjugated payload compared to the lysine-conjugated cargo, the methionine of the FC part in the former sample was oxidized four times more than its latter counterpart [109].

#### 3.2.3. Disulfide Exchanges

It has been observed in many studies that the un-paring and miss-assembly of disulfide bonds in IgG1 can occur in a variety of production processes. One of the earliest stages that initiates this error is the lack of a proper folding system in the endoplasmic reticulum during its production (fermentation) in eukaryotic expression systems [110]. In the case of this antibody, four disulfide bonds between heavy and light chains, as well as two heavy chain disulfide linkages, were found to be more vulnerable than 12 disulfide bonds within the IgG domains [111]. Amano et al., [19] showed that several months need to generate hinge-region cysteine racemization in IgG1, while high pHs and temperature through numerous pathways in a shorter period ultimately can modify the antibody in the same way.

#### 3.2.4. Deamidation

This reaction, which mostly occurs in amino acids Asn and Gln, leads to the transformation of the side chain amide linkage into a carboxyl group [112]. The deamidation of Asn, which mostly occurs in the sequence of Asn-Gly and Asn-Ser, is seen to be more accelerated in neutral and/or alkaline conditions [112]. It has even been understood that the rate of deamidation in these sequences is higher than in free Asn and Gln residues. This shows that the rate of deamidation may be different in the different sites of a protein. Sometimes, if these sequences are in an important position, such as the active site of a pharmaceutical enzyme, it causes the loss of enzyme activity, and if this position is in the structure, it leads to a change in structure, without having a strong effect on the activity. As a result, the consequences of deamidation are different for each protein. By and large, the reaction eventually introduces a negative charge to the protein, which can have different effects on the behavior of the protein. It has been realized that deamidation changes the binding of hemoglobin to oxygen in the sequence of Asn-Gly [113]. It has also been understood that the reaction in the Asn25-Gly26 sequence leads to a change in the activity (decreasing effect) of the porcine adrenocorticotropic hormone (ACTH) [114]. Additionally, incubation of human growth hormone in normal conditions such as a neutral environment (pH 7.4) of natural body temperature (37 °C) for two weeks results in the conversion of 37% of the protein into deamidated form, which will result in a severe decrease in activity [112]. In another study, it was concluded that placing the calmodulin protein at 37 °C (pH 7.4) for one month causes deamidation in such a way that eventually, only 10% of the proteins will remain active [115]. On the other hand, there are controversial results regarding the deamidation of proteins and their immunogenic properties. For example, it has been observed that with the deamidation of mouse epidermal growth factor polypeptide, the altered protein is not significantly different from the control sample in terms of immunogenicity [112]. The same result has been obtained in the mode of deamidation of several antibodies [116]. Nonetheless, in the case of gliadin protein, it has been agreed that the reaction will make the protein more immunogenic [117]. In some studies, even in vivo deamidation has been mentioned in some diseases such as cataracts and Alzheimer’s [118,119]. When a protein is placed in a neutral and/or alkaline environments, the Asn sequence with the succinimide ring intermediate is rapidly converted to products such as isoAsp and n-Asp. It is very important to distinguish these two forms since it is believed that the product isoAsp is responsible for changing the behavior of the protein and, as a result, the occurrence of diseases [120]. In addition to the acidic environment and protein sequence, other factors such as the ionic strength, the tertiary structure of the protein, temperature and buffer content can affect the rate of protein deamidation [121]. Accordingly, investigating this reaction not only can be an answer for some neurodegenerative diseases in the living environment, but it may be a mechanism to prevent this unwanted reaction in the industrial environment and the field of production and storage of medicinal proteins and peptides.

#### 3.2.5. Conjugation-Induced Instability

ADCs and some other protein drugs require conjugation reactions in their manufacture process. In addition to carrying toxic drugs, various linkers such as sugars or polymers [122] are conjugated to improve drug efficacy. The manufacture of conjugation, however, requires several specific steps and sometimes at elevated temperatures (typically 37 °C) which may induce chemical or physical changes. Additional buffer exchange, solvent removal, and concentration steps are also involved in the final manufacture and may further accelerate the changes and instability. 

Cysteine and lysine conjugations are so far the most common conjugation residues and both are used an activation step to provide some degree of site-specificity in conjugation. The fabrication of cysteine-linked ADCs uses cysteine conjugation primarily through maleimide-thiol reactive chemistry. Although generally considered to be a chemically stable bioconjugation approach, there is clear evidence that this conjugation is reversible, and this reversibility manifests itself with the concern of free thiols in circulation. For example, the presence of free thiols may induce cysteine racemization in the hinge region of IgG1 19 and CH2/hinge region destabilization is linked to ADC aggregation. The lysine conjugation method involving an activating step with a dual functional linker has also been reported to introduce a risk of cross-linking. It has been stated in the literature that the change may even be in a place far from the active site of a protein, however, eventually the conjugation leads to a decrease in drug activity [123].

It should be noted that there is no general rule to categorize the susceptibility of peptides and proteins to different stresses. For example, the large BSA protein (66.4 kDa), although it is 50% hydrophobic [124] and has 17 disulfide bonds, it is considered a protein resistant to chemical and thermal stresses. Nonetheless, with two disulfide bonds and 22 kDa size, growth hormone has been considered an unstable protein [125]. However, in general, it has been stated that peptides are more sensitive to physicochemical stresses such as temperature instability, microbial digestions, and protease cleavage [1]. Due to fewer intramolecular interactions compared to large proteins, peptides are also vulnerable to those stresses that target non-covalent interactions. Instead, stresses that are directly dependent on the sequence of amino acids, such as light- or chemical-dependent oxidation, glycation, and deamidation may be more dangerous for large proteins due to having more susceptible amino acids. Similar to other characteristics, general principles cannot be stated for agitation-induced aggregation (and hydrophobic susceptibility of proteins and peptides). Depending on the position where the hydrophobic domains of proteins are placed and what percentage of the final structure has an alpha helix or beta sheet structure, the weakness of the target molecule to the damage will be different. Speaking about the secondary structure of proteins, in general, according to the destruction energy of alpha and beta structures, it has been stated that proteins that have a high percentage of beta sheets are more resistant than heta proteins that have a higher percentage of alpha.

As it was implied, the amino acid content of a protein and/or peptide plays an important role in the internal interactions of the protein and finally its resistance level against environmental stresses. For example, most thermostable proteins have both negatively and positively charged amino acids, which give rise to salt bridge and cations π interactions [126,127]. Depending on the distribution of these amino acids in large or small proteins, the resistance of that molecule to environmental stresses (generally caused by temperature) will vary. It should not be overlooked that in larger proteins, the presence of these amino acids may be higher according to statistical principles, but other factors such as hydrophobic domains and amino acids sensitive to deamidation and oxidation are also higher.

## 4. Strategies in the Stabilization 

In many cases of the delivery of proteins and peptides for research use only, the powdered/dried form of proteins is preferable to their soluble form. In this delivery route, there is no protein hydrolysis pathways, in-solution decomposition, or air–water interactions caused by the agitation of the protein solution and environmental pH changes. Nevertheless, for the treatment sector and therapeutic usage, the powder form of protein drugs may cause many mistakes and consequences for patients, including mistakes in determining the correct dose of the drug, using solutions that may not have the required standard, and/or the need to send approved solvents from the companies. Thus, it is better to think through the solution form of protein and peptide drugs for therapeutic purposes (that is currently being considered) [10]. Some of these strategies are related to the genetic design to produce more stable protein analogs [128], and some other paths are connected to protein structure and solvent engineering [129]. The changes in genetics and the surrounding environment of medicinal proteins can be done with various goals such as increasing activity, increasing solubility, and protein stability. Figure 6 provides an overview of various approaches of the latter aim to stabilize pharmaceutical proteins.

Some of these approaches are related to the primary protein sequence (genetic changes) and many of these strategies are linked to protein changes after production and purification; the latter class usually has been divided into two main sub-classes: non-covalent and covalent modifications.

### 4.1. Genetic Engineering: Protein Analogs

With the discovery of genetic engineering and gaining the knowledge that it is possible to change the DNA sequence in our favor, in 1973, the hope was revived to start producing proteins with properties beyond the natural states. From that year onwards, precisely 1982, and with the creation of the first protein drug with a human genetic source (insulin), it was gradually added to the number of recombinant drugs [1]. Today, with all types of analog insulins, all types of stability and speed of activity are accessible to diabetic patients [14]. The genetic manipulation of medicinal proteins and peptides should be aimed at not harming the protein’s activity as much as possible and only stepping in the valley of its stability must be careful. Meanwhile, two general strategies for the production of analog proteins using direct mutation (site-directed mutagenesis) and random mutation of proteins and the production of fusion proteins will be discussed.

#### 4.1.1. Site-Directed Mutagenesis

There are countless ways to induce a mutation in the genetic structure of the protein to produce a protein analog, which is to create a more resistant or even more active species than the wild form of the protein. For the purposeful creation of protein analog species, one must have sufficient knowledge of the structure of the wild-type protein. Among all changes that lead to the production of analog or mutant proteins, it is worthwhile to refer to substitution, deletion and addition of one or more amino acids in the protein structure. As an example, in our previous study, two amino acids lysine were added at the end of the B chain of insulin, which led to the aggregation of obtained analogs during subcutaneous injection, which increased the in vivo stability of these insulins [130]. Besides, sometimes, some essential amino acids are also added to native proteins to introduce the glycosylation and acylation sites, for increasing the biological and physical stabilities of the target proteins with the help of sugar and fatty acid molecules.

Studying some successful protein drugs such as interferon (IFN) β1b, human fibroblast growth factor (FGF) and granulocyte colony-stimulating factor (G-CSF), has had the result that the substitution of cysteine residue with serine has led to the production of some analogs with a longer half-life [131,132]. Today, with the help of in silico platforms for estimating and predicting domains sensitive to aggregation (such as hydrophobic domains buried in the structure of proteins), researchers can succeed in mutations that make the resulting proteins more stable against temperature and other physical stresses. Human growth hormone (hGH) and G-CSF can be mentioned among these case studies. In the first case, it was reported that the obtained hGH analogs with 10–14 mutations showed a four- to ten-fold increase in shelf life and an increase in temperature stability. Speaking about G-CSF, it was also reported that in addition to an increase in temperature stability up to 16 °C compared to the wild-type protein, the analog has activity in cell proliferation studies, as well [133,134]. As previously reviewed, the digestion/degradation and cleavage of medicinal proteins in many chemical modifications such as oxidation and deamidation can be the coup de grâce to the proteins. Accordingly, efforts to create mutants resistant to protein digestion can be valuable. For example, it is commonly seen that the site of proteolysis of proteins is evolutionarily institutionalized in the flexible loops. Thus, mutations that increase the flexibility of proteins may cause resistance to proteolytic digestion [134]. As an example, it has been found that the mutation of two arginine amino acids in the flexible region of factor VIIIa protein leads to the resistance of this medicinal protein to the proteases of thrombin, activated protein C, and factor VIIIa [135].

#### 4.1.2. Fusion Strategies 

Usually, the most important purpose of creating a fusion state in medicinal proteins is to increase their half-life in the blood circulation system and significantly improve pharmacokinetic profiles [136]. For this reason, it is often used to increase the size of protein molecules to escape from the renal filtration system. In this achievement, a protein that can circulate in the blood for a long period is usually used (such as albumin or the FC part of antibodies) [137]. As a famous and appreciated example, the etanercept protein that is used medicinally for rheumatoid arthritis disease, today, is obtained from the addition of the extracellular domain of p75 tumor necrosis factor receptor (TNFR) and the Fc part of IgG antibody. The FC part leads to an increase in the life of the protein in the blood circulation by the mechanisms of increasing the size of the fusion protein and possibly by endosomal recycling. On the other hand, because the Fc part is a dimer, the binding rate of this protein to TNF-alpha is between 50 and 1000 times higher than that of monomeric TNF-alpha samples [134]. Regarding albumin, two major mechanisms have been proposed. One is the direct binding of this protein to the medicinal proteins (e.g., different variants of barbourin [138] and hirudin [139] and the other is the binding of peptides that can act as linkers between the medicinal protein and albumin (peptides that have a high binding tendency to albumin); extending an albumin-binding peptide label to the anti-tissue factor D3H44 Fab that increased the half-life nearly 40-fold [140]. In addition to these two natural protein molecules, other molecules have also been used to produce fusion therapeutic proteins. In a general view, the following table (Table 1) refers to the types of different fusion partners that have been used to produce fusion proteins.

Sometimes, the purpose of creating a fusion is to increase the stability of the medicinal protein in bacterial expression systems. Especially for therapeutic peptides, the expression of such small molecules in prokaryotic systems such as *E. coli* is always accompanied by the challenge of protease digestion in the bacterial cells, which leads to a low yield of the final product. On the other hand, by designing fusion (carrier protein) to the target peptide, post-expression pathways such as purification will take place more easily. As a previous attempt, in the expression system of *E. coli*, the genes of insulin A and B chains were separately fused with the chaperone protein carrier alpha-B crystallin of the human eye lens, in addition to increasing the efficiency of the expression in the bacterial platform, the purification pathways were also conducted more simply [141]. Recently, a similar study has been carried out to produce proinsulin in the prokaryotic system [142].

### 4.2. Covalent Engineering 

In this section, the purpose is to explain those changes that are covalently applied to the structure of proteins and peptides to increase their stability. In general, this category of changes includes the connection of synthetic polymer molecules such as polyethylene glycol and some other extracted molecules: sugars and fatty acids.

#### 4.2.1. Protein–Polymer Conjugates

The challenges to the stability of medicinal proteins and peptides are so deep that they cannot be completely solved by the strategies reviewed so far. In many cases, low-stability proteins need to be conjugated with synthetic polymers to brighten the hope of increasing their stability. One of the most prominent examples of protein-polymer conjugation is the PEGylation of medicinal proteins, which leads to an increase in the life of proteins in the circulatory systems [143]. There are currently more than 40 FDA-approved protein drugs that have been conjugated by PEG [144]. In addition to increasing the life expectancy in plasma, other benefits such as protein stability in the digestive system, control of receptor-ligand binding, and improving the storage stability of medicinal proteins using PEG polymer have been reported. It is necessary to take great care about the selection of the type of protein, the type of polymer and even the conjugation method, to follow the paths that the guest protein does not change from its natural state (fold) to an unnatural form (unfold) [143].

In this section, the types of polymers that have been used for the engineering of medicinal proteins will be discussed. In order to correctly and systematically choose a suitable polymer for conjugation to protein, it is necessary to consider several criteria, which are discussed in the table below (Table 2). 

Additionally, in the selection of protein, attention should be paid to the types of functional groups that can react with the selected polymer. These functional groups should not be considered in the sensitive areas or the active site of the protein. The following figure (Figure 7) shows the types of possible active groups in proteins.

In addition to both C- and N-ends of a protein, disulfide bonds, side chains of lysine and tyrosine as well as free thiol can undergo purposely various chemical changes to increase protein stability.

The suitable environment for the reactivity of lysine (pKa~10.5) is more in the neutral to basic ranges, which is not suitable for some proteins that are not stable in this range [145]. Additionally, the amino acid surface accessibility and the charge around it will be effective in its reactivity. In the case of cysteine, because it is less abundant than lysine in the primary structure of proteins, the residue is more popular for targeted chemical modifications. Nevertheless, the changes in this amino acid are a bit riskier and can lead to changes in the protein structure. So, the structural characterization of proteins should be investigated after cysteine modification. In many medicinal proteins and especially antibodies, there are a large number of disulfide bonds, which with their slight reduction, it has been seen that it does not cause severe damage to the tertiary structure of the proteins. The same method is used for the changes of some proteins in the reduced cysteines. For example, in the brentuximab vedotin antibody model, after partial reduction of the disulfide bonds, the obtained structure is then conjugated to the maleimide functionalized anti-cancer drug [145]. This method cannot be useful for small proteins with a small number of disulfide bonds, since the whole structure may collapse by reducing the bonds. For chemical changes in the disulfide bonds of these proteins, replacing natural disulfide bonds with synthetic ones has obtained hopeful results [146]. Additionally, pka for the N-terminus amine of proteins is usually reported 6–8, which is usually used as a mindset for changes in the amine end (pH-dependent reactions). For example, the pegfilgrastim protein drug with the same method has a PEG at the N-terminus amine. Reductive amination in acidic pH usually leads to N-terminus amine PEGylation of proteins with yields above 90% [145,147]. However, it should be kept in mind that connecting the polymer to the N-terminus amine will be advantageous in proteins that are small and have a few lysine residues. Because the amine group located in the side chain of lysine can win by competing with the amine at the end of the protein and reducing the percentage of the final product. Usually, it is difficult to distinguish between the C-terminus of the protein and the carboxyl present in the side chain of amino acids Asp and Glu, which indicates the necessity of using tags at the C-terminus to confirm the reactivity of this end [148,149]. Tyrosine is also a residue that has a low abundance on the surface of proteins and is usually not followed by high popularity in the field of polymer-protein conjugation due to its cross-reactivity with the amino acid histidine in nucleophilic reactions.

#### 4.2.2. Linker Chemistry

The linker is a very essential part of protein conjugation, especially for ADCs. Linker chemistry can be tuned to impart adequate stability for protein drugs [150]. As mentioned above, PEGylation is an example of using hydrophilic PEG linkers to improve drug efficacy [151]. There are many factors to consider when choosing linkers for ADCs. Nevertheless, one of the most important of them is the stability of the linker throughout its presence in the plasma, while it must break down as soon as it enters the target cell (in the form of a conjugate with ADCs) and leads to the activation of the released anti-cancer drugs [152]. There are cleavable and non-cleavable linkers. The cleavable linkers are cleaved by chemical or biochemical changes in the environment such as acidic changes for hydrozone linkers and glutathione or high pH values for disulfide linkers. Some other cleavable linkers contain enzymatic digestion sites and are sensitive to the enzymes inside the vesicle such as valine-citrulline di-peptides for cathepsin B enzyme. On the other hand, the non-cleavable linkers depend on the complete digestion of the antibody after entering the vesicle [153]. 

Among the important reasons that led to the removal of Mylotarg from the marketing list in 2010 were items such as the instability of the hydrazine linker and its payload release [154,155]. More recently, Trastuzumab-emtansine (T-DM1) 1 with a non-cleavable linker is the first generation of ADCs approved for the treatment of HER2 (human epidermal growth factor receptor 2)-positive breast cancer. Second-generation ADCs targeting HER2, trastuzumab duocarmazine (SYD985), was characterized to supersede T-DM1 by overcoming all resistance using a cleavable linker and a more potent payload, duocarmycin (DUBA) via different conjugation chemistry. There are, however, no measurable changes observed between the half-life of Adcetris^®^ with a cleavable linker and Kadcyla^®^ with non-cleavable linker at the clinical level [156,157]. 

The linker property will also determine how the final products act during blood circulation and also touch the sensitivity of the antibody to environmental stresses [158]. Some disulfide-linked ADCs were found to conjugate with blood proteins through cleaved free thiols. Hydrophobic linkers and payloads frequently encourage the aggregation of ADC molecules, e.g., King et al. showed that multiple loading of the BR96 antibody that is simultaneously attached to doxorubicin drug leads to non-covalent dimerization of the antibody, which will eventually lead to hepatotoxicity [159,160]. It has also been seen that the linkers that are more hydrophobic are not able to significantly affect the MDR1^+^ cells. Instead, linkers with a more polar and amphiphilic nature were able to appear with a greater effect such as mal-PEG4-N-hydroxysuccinimide and N-Hydroxysuccinimidyl-4-(2-pyridyldithio)-2-sulfobutanoate (sulfo-SPDB) [161]. In addition to the MDR1 challenge, it has also been found that reducing the percentage of hydrophobicity by employing hydrophilic linkers containing negatively charged sulfonate groups, polyethylene glycol (PEG) groups, or pyrophosphate diester groups can also help in improving therapeutic index level and pharmacokinetics [162]. 

#### 4.2.3. Acylation 

This strategy has mostly been used in the case of medicinal peptides; liraglutide and insulin detemir are among the most successful cases of lipidated peptides in the world market. It has been reported that lipidation in the backbone of peptides can give positive characteristics to the obtained product, such as impressive stability against enzymatic digestion, and specificity to the receptor and also to the bioavailability of peptides. In addition to the issues related to the stabilization of medicinal peptides, this process has received much attention in the field of drug delivery of medicinal peptides [163]. This process occurs unsurprisingly in nature. In the living system, various enzymes help a protein or peptide to react with a chain of saturated or unsaturated fatty acids in the form of N- and/or O-links [164]. The binding site of lipids can be the terminal amine and carboxyl, and even in the middle part of the primary protein/peptide structure, which occurs through residues such as serine, threonine, glycine, lysine, and cysteine [165]. It is interesting to note that this addition of lipids can be irreversible (when it is through the connection of the N-terminal of glycine/cysteine) or reversible (when a thioester bond occurs between the fatty acid and thiol group of cysteine) [166,167]. Regarding the medicinal aspects of lipidopeptides, some of these peptides are naturally extracted from microorganism sources such as fungi and bacteria. The use cases of this type of peptides have been proven in anti-cancer, anti-fungal and anti-bacterial fields [2]. However, from another line of view, in silico addition of lipids to peptides is also of special interest [168]. In addition to the fact that the presence of peptides can lead to the appearance of some new features in the peptide, it is reported that the type of spacer between the lipid and the peptide, the type of lipid and its chemical nature (saturated and unsaturated) are also effective in the activity of the lipidopeptide [169,170]. The table below (Table 3) mentions some of the peptides that have been joined with lipids as encouraging options for treatment.

#### 4.2.4. Cyclization

Cyclization is another solution that has been proposed to increase the stability of medicinal peptides, especially against enzymatic digestions. The idea of this strategy is taken from nature by discovering peptides such as mammalian theta-defensins [180] and plant cyclotides [181]. The best example of an engineered cyclic peptide is a 29 amino acids peptide, kalata-B1, which is resistant to a wide range of environmental stresses such as 8 M urea, acidic environments (0.5 M HCl), 6 M GnHCl, and the presence of various proteases [182]. Taken as a whole, the high stability of these cyclic peptides has caused special attention in recent years to search for faster synthesis methods and the purification of cyclized peptides.

At first, in the following table (Table 4), several circular medicinal peptides will be introduced, and then, due to the importance of this section, various methods of peptide ringing will be discussed. As can be deduced from the information in the table, most cyclic peptides belong to non-synthetic peptides, extracted from microorganisms. This is due to the complex routes of cyclization of peptides and finally their purification to an acceptable level [183].

Based on the nature of the linkage between the two residues in the peptide structure that ultimately ensures the cyclicity, there are two major classes of cyclic peptides: homodetic (involving only a conventional peptide bond) and heterodetic (using a variety of functional groups that required for looping the peptides) [208]. 

#### 4.2.5. Nanoparticles; Double-Blade Sword

In most past studies, nanoparticles have been used as a platform for the delivery of protein and peptide drugs. The oral route of using biological agents is the easiest way to use such drugs, but due to the destructive conditions of the digestive system (acidic environment, the presence of many protease enzymes, and a small percentage of surface absorption in the small intestine), this route has always been associated with many challenges. For this reason, the use of nanoparticles has been focused on facilitating the delivery of protein drugs in the alimentary tract. However, in some studies, nanoparticle systems have also been used for the intravenous routes of drugs. In any case, for the use of nanoplatforms for the delivery of protein or peptide drugs, the surface and nanoparticle size are factors that should be taken into consideration [209]. From the surface chemistry point of view, firstly, nanoparticles should ideally have hydrophilic surfaces that can be easily dissolved in living systems and secondly escape from the macrophage system [210].

Given particle size, in living systems of vertebrates, macrophages are responsible for ingesting large foreign particles (size around 0.5 µm). The selected nanoparticles should be small enough to escape from this system on the one hand, and on the other hand, they should be large enough to not quickly enter the outside environment of the vessels through the vascular leakage system. It has been agreed that this size is between 150–200 nanometers in the spleen and liver on the one hand, and on the other hand, the gap junctions between endothelial cells are between 100 and 600 nanometers [211,212]. According to this evidence, choosing the right size is a challenging factor. However, looking at the successful studies in this field, in general, it is understood that the useful nanoparticles that have been used so far have been between 3 and 200 nanometers in size. A wide range of nanoparticles have been used to deliver all kinds of drugs, nevertheless, in the case of protein and peptide drugs, most of the attention has been on polymer nanoparticles and solid lipid nanoparticles. Of these two groups, the latter has also been pursued more seriously due to the controlled release of the cargo and its ability to be engineered for a specific organ [213]. To summarize the contents of this section, it is necessary to mention the following table (Table 5), which presents the types of nanoparticles, the cargo, and the delivery method used in previous years.

### 4.3. Non-Covalent Engineering of Therapeutic Proteins and Peptides; Solvent Engineering Pathways

#### 4.3.1. Formulation: Solvent Engineering Pathways

Another solution that has been used to preserve the integrity of medicinal proteins and peptides is the use of excipients during storage and even through the use of such drugs. According to the types of these agents, their mechanisms of action are also diverse [228]. In the following table (Table 6), the types of these excipients are presented along with their mechanism to protect proteins. Some of these excipients should be used for a certain amount of window with specific considerations such as salts that can lead to the aggregation of proteins after a critical concentration [229]. Or as another important example, in the category of sugars, usually non-reducing sugars are used because reducing sugars via Millard’s reactions may cause destructive glycosylation of proteins [230].

According to all the information that was reviewed about the stabilization of proteins and peptides, at the end of this section, it is necessary to briefly suggest which types of proteins usually suffer from which stress and what are the proposed solutions to overcome the challenges (Figure 8).

#### 4.3.2. Choice of Container

Depending on the type and even the architecture of the protein containers, proteins may react with the containers’ surfaces, be adsorbed without or with a change in the tertiary structure, and lead to protein aggregation that is either reversible or irreversible. For example, it has been detected that rhFVIII proteins undergo tertiary structure changes in contact with hydrophobic silica surfaces, and changing the nature of the surface to a negative charge has led to a slight change in the protein structure [233]. It has also resulted in some cases where no significant structural change occurs in the protein, but a severe aggregation behavior occurs during protein binding to the surfaces. For instance, rhPAF-AH protein does not show a measurable change in contact with the hydrophilic surface of silica, though, at a pH 6.5, it showed a very high aggregation rate [234].

Based on past efforts in this field, scientists have generally agreed that glass containers are not ideal, but suitable, options for transporting and storing these therapeutic macromolecules [235,236]. Some studies have also pointed out that even these glass containers can interact with proteins and cause structural changes in them, which in itself can reduce the stability of the protein and other consequences such as ectopic responses in the patient’s body [236]. It is worth mentioning a few studies on the changes that have been made to the surfaces of glass containers to help stabilize drug proteins. From about 1850, therapeutic glass containers were introduced to world markets. Then, in the 1950s, due to problems such as the reactivity of pharmaceutical proteins and some other chemicals with glassware, efforts began to engineer glassware using beneficial polymers [237]. Although glass vials are mainly used for the storage of medications, in some of them, the presence of glass particles, separated from the bottom of the container, has been reported [117]. Container delamination is an undesirable process in which a thin layer of glass is separated from the main body glass and the separated particles have the potential to destroy the drug [238]. For such cases, the US Pharmaceutical Association (USP) has guided assessment glass delamination called USP <1660> [237]. USP <1660> lists most of the possible factors: the composition of the glass, the conditions under which the container is made, the subsequent displacement of the container and the medicinal product in the container. Thus, assessing the inner surface of glass vials, including examining the surface of the glass using an electron microscope (to analyze the degree of the surface cavity or depth), measuring the total composition that may have been separated from the glass using inductively coupled plasma-mass spectrometry, or examination of visible particles using dynamic light diffraction or scanning electron microscopy-energy scattering (SEM-EDX) is essential [237]. Combining all mentioned studies, for industrial technology, glass vials are preferred over plastic vials due to the standard depyrogenation, operations of washing and possibly terminal autoclaving and in another view, glass containers have less immunogenicity in comparison with plastic counterparts [239]. Although in recent years, different particles with an outer layer of silica have also been invented, which can be an alternative option for current protein drugs containers. However, it is still necessary to go through several confirmation stages to industrialize these candidates, [240,241]. 

## 5. Different Approaches in Characterizing Protein Drugs and Induced Structural Changes

The characterization of the chemical and physical stabilities of medicinal proteins is vital in drug development and must directly link to a number of critical quality attributes (CQAs) that must be checked to guarantee final-product quality [242]. Advances in analytical techniques have given researchers and biopharmaceutical companies new and increasingly sensitive methods for characterizing their products. Due to the complexity of pharmaceutical proteins, it is regularly essential to study the stability of them using several techniques [243] based on their changes in mass/size, folding, charge, and hydrophobilicity, as well as optical and thermodynamic properties as drawn in Figure 9.

Although the first sign of the validity of the protein may be its transparent state and its solubility, which can be easily seen even with the naked eye [244], most CQAs require more sensitive monitoring at the molecular level which needs to be examined by analytical instruments. Using LC-MS, subunit and intact protein masses can be directly measured to confirm product identity and quantify different sub-units, as well as to monitor conjugated species and the drug-to-antibody ratio of ADCs at the same time [245]. Moreover, a bottom-up approach relying on the enzymatic digestion of the protein into peptides before LC-MS is used to identify chemical modifications such as oxidation and deamidation on a specific residue site as well as to estimate their modification levels [246]. Due to the varied range of post-translational modifications and different steps during industrialization, pharmaceutical proteins are severely heterogeneous and need high-resolution mass spectrometry (HRMS) for identifying, characterizing, and monitoring these attributes [247]. Disulfide linkages can also be solved by bottom-up LC-MS approach with non-reducing digestion using gas phase [248] and/or solution phase [249] fragmentations. However, bottom-up processing can result in considerable information loss or errors due to mis-cleavages or data mismatching. Combining intact, middle-up, and bottom-up approaches of LCMS or capillary electrophoresis-MS (CE-MS) appears to be one of the main platforms in characterizing and monitoring the chemical stability of pharmaceutical proteins [250]. 

The challenge of protein aggregation has always been discussed to a high degree in industrialization. Regarding this, the size exclusion chromatography (SEC) technique has been known for years, however nowadays, due to the innovation in the field of monitoring protein aggregates, many institutions request orthogonal analytical methods for the accuracy and reliability of the SEC data [251,252]. In addition to SEC, protein aggregation can be detected in various ways such as small-angle X-ray scattering and dynamic light scattering. Fortunately, during the structural changes of proteins and their folding, many signals can be monitored to ensure whether the protein in question is folded or not. Sometimes the size of the protein is considered, in some cases the tertiary structure or the chemical composition of the protein will determine the amount of damage done to the protein. The size, changes in the three-dimensional shape of a protein, and changes in its chemical nature are among the general criteria that can be considered [253]. Native MS, a new technique to simultaneously determine molecular weight and detect the exposed charge profile of a native protein, has been applied to study protein folding and stabilities by varying temperature [254] with unparalleled sensitivity, dynamic range, and selectivity for studies of both cold- and heat-induced chemical processes.

There are many challenges in determining the properties of protein drugs and especially ADCs whose synthesis normally leads to heterogeneous drug loads depending on the kinetics of conjugation chemistry and linkers [255]. Consequently, it is necessary to make a very good purification of ADCs with a specific drug-to-antibody ratio [256]. Hydrophobic interaction chromatography (HIC) has emerged as a central technique for separating protein conjugates with different drug loads for cysteine or related conjugates where the linker-payload molecules add appreciable hydrophobicity to the ADC. Several orthogonal separations techniques have been developed to purify ADCs with different drug loads. For example, two-dimensional liquid chromatography (2D-LC) via two different separation techniques such as ion exchange followed by a reversed-phase was used to intensify the separation influence [257]. The ultra-high voltage capillary electrophoresis (UHV-CE) technique, which usually uses at least 120 kV and an electric field over 2000 V/cm, was initially used to separate peptides and sugars, nevertheless, recently it has also been used to separate ADCs based on their charges [258]. 

Likewise, sometimes the dynamics of proteins become important. In the dynamic environment of the cell, proteins do not exist in a static physical form and are constantly undergoing structural changes to fit and bind to their receptors. Instead, some proteins with oligomers with larger structures can be present in the environment, and techniques such as X-ray crystallography and NMR are not able to monitor such changes. Hydrogen deuterium exchange MS (HDX–MS) [259,260] or other MS-compatible chemical labeling methods such as hydroxyl radicals [261] or reductive amination [262] have been used before MS detection to reveal solvent exposure domain in a protein since these regions are more amiable to chemical instability. Although these techniques require a lot of time to analyze the data, they give a good experimental view of molecular dynamics. To summarize all the contents in this section, the following table (Table 7) tries to categorize the various techniques and methods of analyzing the structure of pharmaceutical proteins coherently. Many of these applied techniques have international standards from authorities such as the World Health Organization (WHO), the Food and Drug Administration (FDA) and the National Institutes of Health (NIH) to determine the accuracy of size [10]. Additionally, these techniques must be evaluated and validated from the aspects of linearity, specificity, limit of detection, precision sensitivity recovery rate, and accuracy [263].

## 6. Conclusions and the Future of Protein and Peptide Drugs

Looking at the sales share of protein and peptide drugs in recent years and their fast-growing trend, it is not far from expected that these macromolecules will cover most of the future therapeutic markets. Considering the positive features of this class of drugs compared to conventional chemical-synthetic drugs, such as high specificity, reasonable biological lifetime and low toxicity in the body, as well as the need for very low doses to create therapeutic responses, large pharmaceutical companies are willing to invest in this field. However, the problem with these drugs, since the beginning of their arrival, is their instability in different environmental conditions. As mentioned, many efforts have been made to solve this problem, but so far no promising strategy has been achieved in this field. For some therapeutic proteins (such as insulin hormone analogs), the binding of chemical molecules such as fatty acids has been a good option to increase their stability. Similar to the new approach used to produce antibody drug-conjugated, it is expected that a similar strategy will be adopted for some other therapeutic macromolecules such as DNA or RNA molecules in the future. Besides, some efforts have been focused not on the structure of the proteins, but on their storage containers. Nowadays, most of the containers for storing therapeutic proteins and peptides are made of borosilicate glasses, which bring challenges such as the crushing of microscopic pieces of them in the solution and diverse interactions of proteins with their surfaces. In the future, it is expected that this part of science will also reach its evolution, meeting the need for stable medicinal proteins. Until now, a huge part of the application of nanoparticles and polymers has been used in the direction of drug (protein/peptides) delivery, however, in the field of engineering containers for medicinal proteins, no coherent effort has been made.

As reviewed, many pathways threaten the folding and shape of proteins/peptides. Likewise, fortunately, many solutions have been implemented to stabilize these important medicinal macromolecules. Consequently, it should be noted that the choice of which stabilization path to use is strongly related to the type of protein/peptide and even the route of administration of the protein in the body. Furthermore, sometimes it is necessary to apply two or even three strategies in combination to a protein to achieve the desired stability. Nevertheless, whatever the plan of stabilization is, it must ensure that the resulting protein remains non-immunogenic and active. Thanks to bio-analytical chemistry, various methods have been obtained to confirm the final structure of the engineered protein/peptide.

## Figures and Tables

**Figure 1 pharmaceutics-14-02533-f001:**
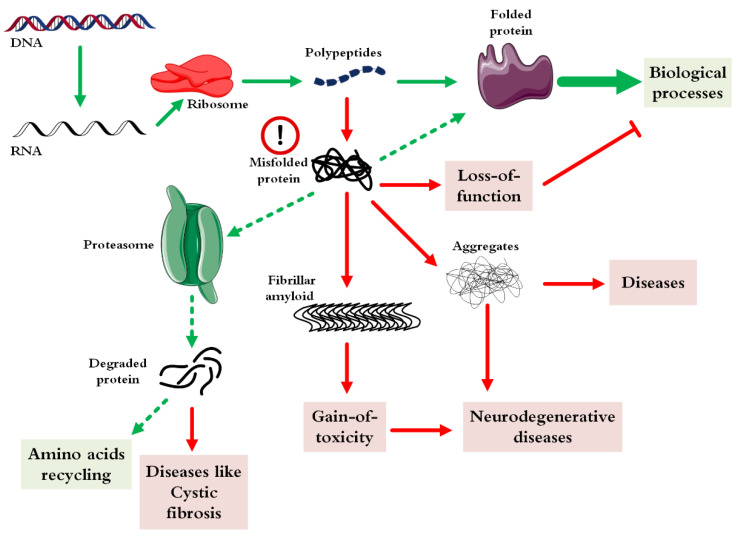
Possible intracellular pathways of correct and/or incorrect protein folding.

**Figure 2 pharmaceutics-14-02533-f002:**
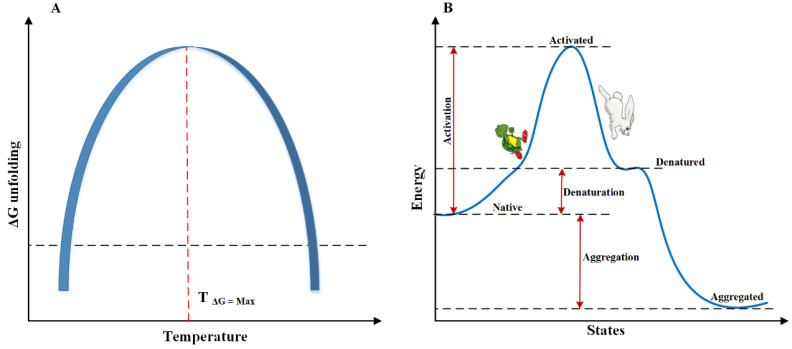
The relationship between protein unfolding and internal energy (the data were extracted from [21,38]). (**A**) Shows the possible quantities of ΔG_unfolding_ at different temperatures. (**B**) Indicates the several stages of the protein unfolding pathway. At first, by obtaining energy, the protein is activated to higher levels by increasing the amount of internal energy, and at a critical point, it unfolds by releasing energy to reach a more stable state. In a large number of studies, it is believed that the most stable state of a protein is its aggregated form (fibril form), however, some researchers believe that its natural form is the most stable state.

**Figure 3 pharmaceutics-14-02533-f003:**
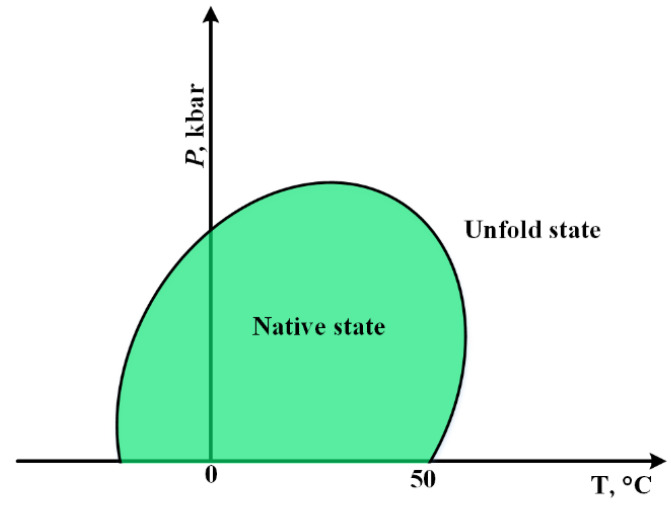
The relationship between pressure and temperature during the folding and unfolding of proteins [54].

**Figure 4 pharmaceutics-14-02533-f004:**
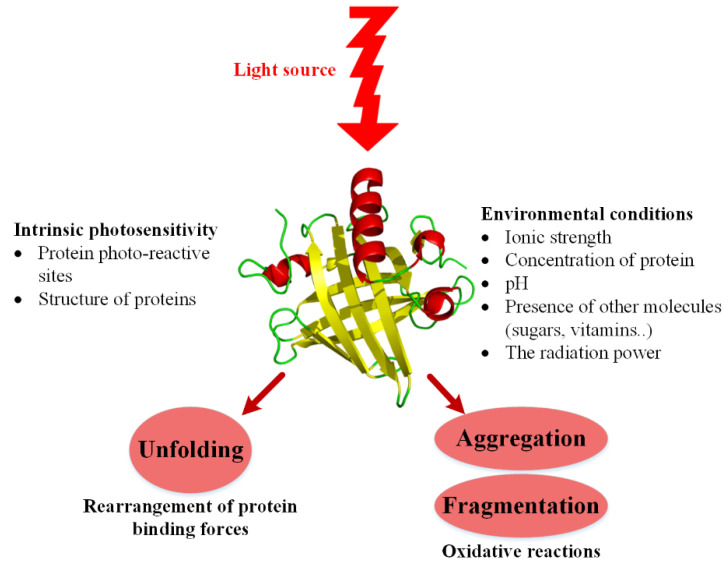
Determinants of photo-susceptibility of proteins (the data were extracted from [63]).

**Figure 5 pharmaceutics-14-02533-f005:**
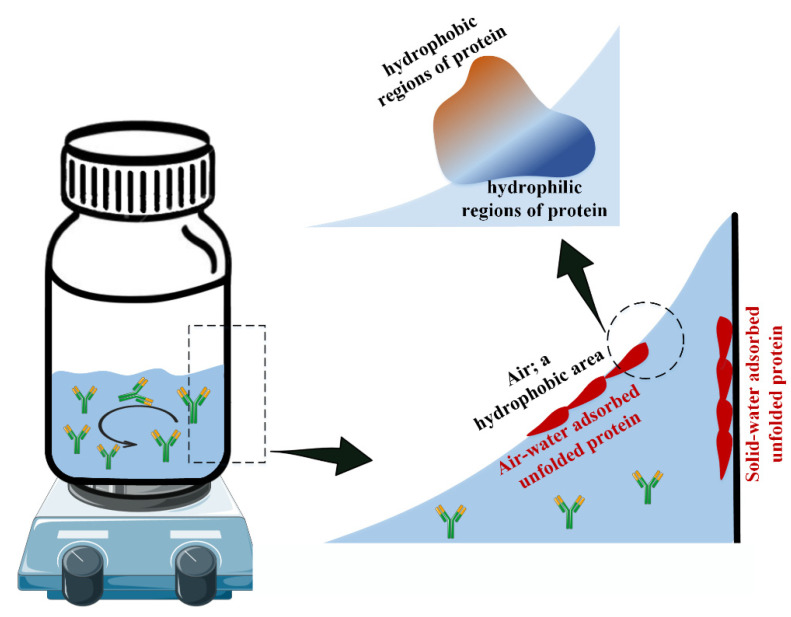
Agitation-induced protein unfolding/aggregation (the concepts were inspired by [79,80]).

**Figure 6 pharmaceutics-14-02533-f006:**
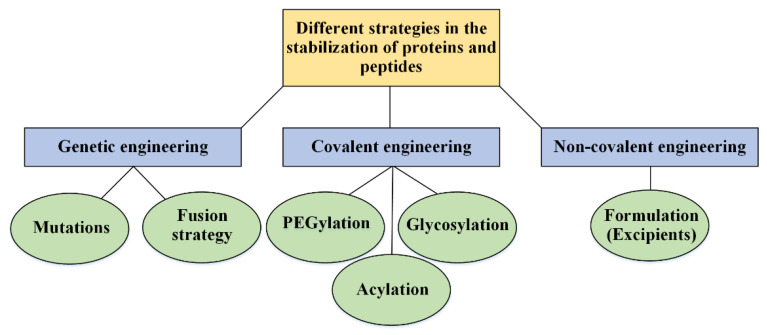
A variety of strategies to increase the stability of medicinal proteins and peptides.

**Figure 7 pharmaceutics-14-02533-f007:**
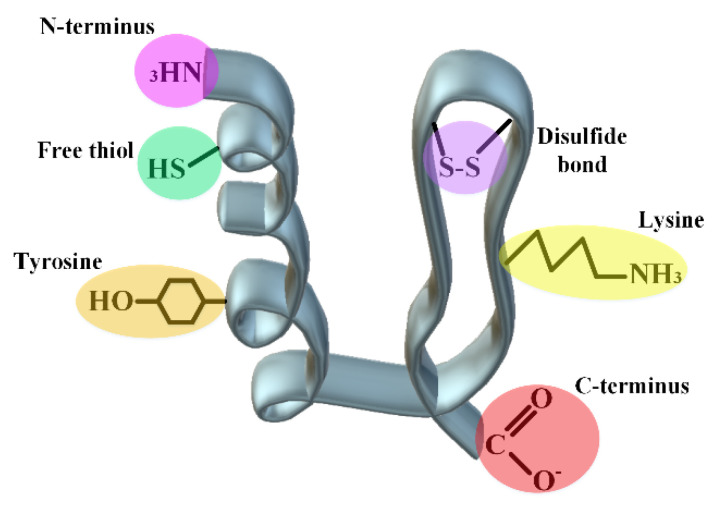
Introducing several active groups on the surface and inside a hypothetical protein structure to introduce desired covalent modifications.

**Figure 8 pharmaceutics-14-02533-f008:**
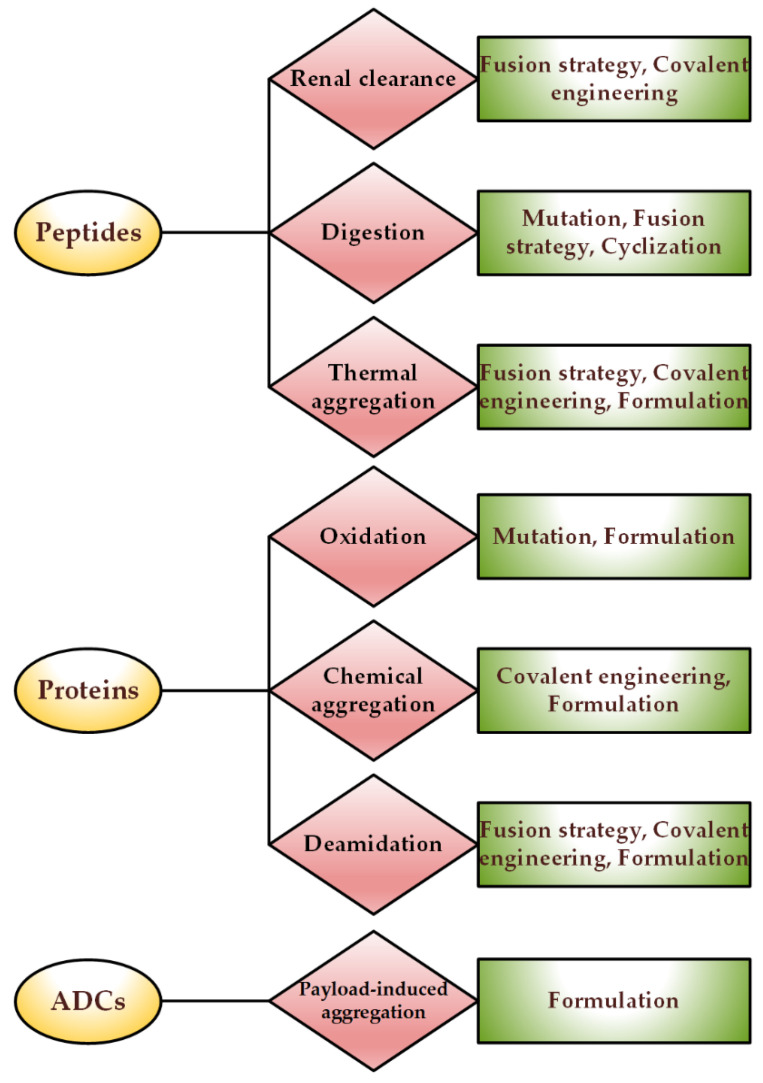
Proposed solutions to the challenges faced by medicinal proteins and peptides.

**Figure 9 pharmaceutics-14-02533-f009:**
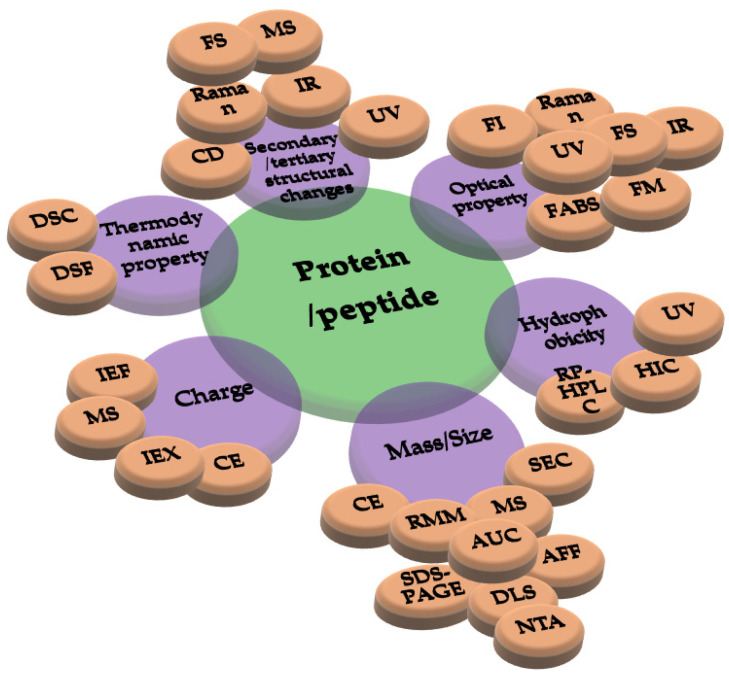
Different analytical techniques that are currently applied for characterization of protein and peptide drugs. Abbreviations: AUC: analytical ultracentrifugation, DSC: differential calorimetry, CE: Capillary electrophoresis, CD: circular dichroism, DSF: differential scanning fluorimetry, UV: ultraviolet, DLS: dynamic light scattering, FAPS: fluorescence-activated particle sorter, AFF: asymmetrical flow-field flow fractionation, NTA: nanoparticle tracking analysis, RMM: resonant mass measurement, SEC: size-exclusion chromatography, SDS-PAGE: sodium dodecyl sulfate-polyacrylamide gel electrophoresis, MS: mass spectrometry, RP-HPLC: reversed-phase high-performance liquid chromatography, IEF: isoelectric focusing, IEX: ion exchange chromatography, FM: Fluorescence microspcopy, FI: Fluorescence imaging, and FS: Fluorescence signal.

**Table 1 pharmaceutics-14-02533-t001:** The introduction of different partners in the fusion proteins production [137].

Partner Type		Marketed Examples	The Effects of the Partner
Non-peptide partners	Fc	Enbrel^®^, Eloctate^®^ and Alprolix^®^ (FDA approved).	Enlargement to a size greater than 70 kDa (the critical size to escape renal excretion).
	Albumin	Tanzeum^®^ (FDA approved).	Increases blood circulation
	Transferrin	-	Improve oral administration, no FDA-approved cases,
	Carboxyl terminal peptide	Elonva^®^ (produced by Merck and Co., Rahway, NJ, USA, but not FDA approved). Lagova^TM^ (in phase 3 clinical trials).	31 residues with four O-glycosylation sites
Recombinant polypeptide partners	XTEN	-	Increases blood circulation
	Elastin-like polypeptides	Glymera^TM^, Vasomera^TM^	Contains V-P-G-x-G, the sequence where x is any residue except proline.
	Proline-alanine-serine	XL-Protein GmbH (preclinical state).	The repetition of Pro, Ala and Ser residues in 100–200 copies. Delaying the renal clearance
	Glycine-Serine rich peptides	-	As a linker and a partner for medicinal peptides

Abbreviations: FDA: food and drug administration, Fc: crystallizable fragment, Tf: transferrin, ELP: Elastin-like polypeptides.

**Table 2 pharmaceutics-14-02533-t002:** Essential considerations in selecting a suitable polymer for protein-polymer conjugation [143].

Factors Must Be Consider	Example	Description
Identity	PEG and PEG analogs	Increases the hydrodynamic radius, decreasing immunogenicity
	Stimuli-responsive polymers	Thermo responsive: p(NIPAAm) pH-responsive: poly(acrylic acid) and p(DMAEMA)
	Biomimetic polymers	Trehalose Heparin-mimicking polymers: poly(styrene sulfonate) and poly(vinyl sulfonate)
	Degradable polymers	Cyclic ketene acetals copolymerized with vinyl monomers: degradable under basic pH Hydroxyethyl starch: a-amylase sensitive
Toxicity	Hydroxyethyl starch	Plasma volume expander
Molecular weight	PEGylated proteins	Decreasing renal filtration
Polymer architecture	Branched, brush and linear polymers are accessible to be conjugate with proteins.	Stimulating the immune system
Polymer solubility	-	Soluble in the range between 100 and 500 mg/mL

Abbreviations: PEG: poly ethylene glycol; p(NIPAAm): poly(Nisopropylacrylamide); p(DMAEMA): poly((N,N-dimethylamino)ethyl methacrylate).

**Table 3 pharmaceutics-14-02533-t003:** The list of some peptide-lipid conjugates.

Lipidopeptide/Protein	Lipid	Attachment Strategy	Effects	Ref.
Liraglutide	Palmitic acid	γ-glutamic acid	Extension of the half-life, reducing renal clearance	[171]
Salmon calcitonin	N-palmitoylated	Cys-1 and Cys-7 of the peptide	Extension of the half-life, reducing renal clearance	[172]
Opioid peptide leu-enkephalin	3,4 bis(decylthiomethyl)- 2,5-furandione 16	N-terminus binding	Reducing receptor binding, increasing protease stability	[173]
Thyrotropin-releasing hormone	lauric acid	N-terminus binding	Peptide penetration across the small intestine	[174,175]
H-Ras	1,2-dioleoyl-sn-glycero-3-phosphoethanolamine	maleimide-functionalized phospholipids/ S-palmitoylation at Cys181 and Cys184	Dimerization	[176]
Caveolin-3	octanethiol	cysteine S-fatty acylation	As a small protein model in the study of lipidation of proteins	[177]
Interferon-induced transmembrane protein 3 (IFITM3)	maleimide palmitate	maleimide-functionalized phospholipids/ S-palmitoylation at Cys72 and Cys105	Enhancing the conserved amphipathic domain	[178]
Cathepsin-B inhibitor	Palmitic acid	Palmitoylation of the amino groups (ε-lysyl amino groups)	Increasing the inhibitory effect on cathepsin-B	[179]

**Table 4 pharmaceutics-14-02533-t004:** Some examples of naturally extracted and chemically synthesized cyclic peptides with their applications.

Name	Source	Application	Ref.
Sunflower trypsin inhibitor 1 (SFTI-1)	*Helianthus annuus*	Trypsin inhibitor, angiogenic activity	[184]
Gramicidin S	*Bacillus brevis*	Antibiotic activity towards Gram-negative, Gram-positive and several pathogenic fungi	[185,186]
Plitidepsin	*Aplidium albicans*	*In vitro* anticancer activity	[187,188]
Tyrocidine	*Bacillus brevis*	Antibiotic action	[189]
Rakicidin F	*Streptomyces*	Antibacterial effect	[190]
Plitidepsin	*Aplidium albicans*	Antitumor, antiviral and immunosuppressive activities	[191]
Asperpeptide A cyclo(-Pro-Ala-Ala-Tyr-5-OHAA)	*Aspergillus sp. XS-20090B15*	Antibacterial activity	[192]
Cyclosporin A	*Tolypocladium inflatum*	Calcineurin inhibitor, decreasing the function of lymphocytes	[193]
Depsilipopeptide colisporifungin	*Colispora cavincola*	Antifungal activity	[194]
Cryptophycin	*Nostoc*	Fungicide and anticancer	[195]
Romidepsin	*Chromobacterium violaceum*	Apoptotic activity, anticancer activity	[196,197]
Theonellamide G	*Theonella swinhoei*	Cytotoxic activity	[198]
Ziconotide	*Conus magus*	Analgesic agent; a strong pain killer.	[199]
Kahalalide F	*Elysia rufescens*	Antitumor	[200]
Vancomycin	*Amycolatopsis orientalis*	Antibacterial	[201]
Alisporivir	Chemically synthesized from ciclosporin.	Inhibits cyclophilin A, the potential effect on Alzheimer’s disease and hepatitis C	[202,203]
Istodax (Romidepsin)	Head–tail lactone cyclization which is stabilized by a pair of disulfide bonds.	Anticancer	[204]
Lupkynis (Voclosporin)	Head–tail cyclization	Calcineurin like activity	[205]
Vasostrict (Vasopressin)	Disulfide mediated cyclization	Used in Anti-diuretic hormone deficiency	
Signifor (Pasireotide)	Head–tail cyclization	Activating a broad spectrum of somatostatin receptors, reducing cortisol secretion	[206]
Prialt (Ziconotide)	Disulfide mediated cyclization	Pain killer	[207]

**Table 5 pharmaceutics-14-02533-t005:** Examples of those nanoparticles have been used to stabilize proteins and peptides.

Nanoparticle	Protein Cargo	Delivery Rout	Ref.
Chitosan functionalized lipid nanoparticle (LN)	Insulin	Oral	[214]
Chitosan-modified mesoporous silica	Insulin	Suggested for oral delivery	[215]
Chitosan-modified carboxymethyl-β-cyclodextrin	BSA	Oral	[216]
Poly(lactide)-tocopheryl polyethyde glycol succinate	BSA	Oral	[217]
Alginate	Osteogenic protein-1	Intranasal	[218]
DOPE-liposome	E75 peptide (HER-2/neu-369–377)	Intravenous	[219]
DOTAP-liposome	HSV-derivated E7 oncoprotein	Intravenous	[220]
Chitosan-coated liposomal system	Ghrelin (a peptide hormone that can regulate appetite and body weight changing)	Nose-to-brain delivery	[221]
Chitosan-covered liposomes	Rainbow trout skin-derived peptides	Suggested for oral delivery	[222]
PEGylated liposomes	Opiorphin	Intravenous	[223]
Polyacrylate-coated superparamagnetic Fe3O4	Elastin-like VPGVG pentapeptides	Subcutaneous injections	[223]
G4 OH-terminated PAMAM	AFPQFRSATLLL	Subcutaneous injections	[224]
Thiolated chitosan nanoparticles	Insulin	Subcutaneous injections	[225]
Chitosan-modified PLGA nanospheres	Elcatonin	Pulmonary	[226]
Chitosan nanoparticles	RGD peptide (Arg-Gly-Asp)	Intravenous	[227]

Abbreviations: DOPE: 1,2-Dioleoyl-sn-glycero-3-phosphoethanolamine, HER: Herceptin, GLP-1: glucacon-like peptide-1, DOTAP: N-(2,3-Dioleoyloxy-1-propyl)trimethylammonium methyl sulfate, HSV: herpes simplex virus, PAMAM: Poly(amidoamine), PLGA: poly(lactic-co-glycolic acid).

**Table 6 pharmaceutics-14-02533-t006:** Different types of additives for the formulation of protein and peptide drugs [7,10,231,232].

Excipient	Mechanism	General Comments
Buffering agents	Keeping the pH of protein solutions	Usually, the buffers work in the range of 3–10 for proteins. In certain conditions, some buffers may be decomposed, and their by-products destroy the protein structure.
Chelators and antioxidants	The roles of antioxidants and chelators are to prevent and/or remove oxidazing factors.	Some reducing agents such as glutathione and ascorbic acid in the presence of metals and enhancing oxidation stresses can have a negative/destructive role on protein structure, although these agents are used in the pharmaceutical sector of proteins.
Proteins	By interacting with therapeutic peptides, excipient proteins can increase the blood circulation time.	Nowadays, chaperones have been given a special look as a preservative for medicinal proteins.
Polymers	Maintaining the structure of proteins	The main examples of this group include polyvinyl alcohol, dextran and hydroxyethyl starch.
Amino acids	Buffering properties, preferential interactions, favored hydration, antioxidant effect and strong binding to protein regions	Glycine (buffering agent and bulking agent during lyophilization), arginine (solubilizing agent and works as chaperone) and histidine (antioxidant and buffering agent)
Sugars and carbohydrates	Forming a crystal network with preferential interactions	Sorbitol in lyophilization and liquid formulation conditions, has a stabilizing role, however, in freezing conditions, due to the formation of sorbitol crystals, it assumes a stabilizing role.
Salts	Tonicifying agent, reacting with the charged surfaces of proteins and take two stabilizing or destabilizing paths.	Among the cations and anions of the Hofmeister series, the latter has dual protective/destructive effects on proteins.
Antimicrobial preservatives	Preventing the growth of bacteria in solution-rich medicinal proteins	m-cresol, phenol and benzyl alcohol are among the popular.
Surfactants	Reducing the interface area of solution and air during purification (inner wall of purification column and dialysis bags, etc.).	In this group, polysorbate 20 (PS20) and polysorbate 80 (PS80) are the most used in protein drugs and especially antibodies.
Osmolytes	Generating favored hydration, preferential interactions, and polar interactions	Sorbitol, sucrose, glycine, and trehalose have been able to reach the pharmaceutical sector

**Table 7 pharmaceutics-14-02533-t007:** The list of analytical techniques to characterize medicinal proteins and peptides [39,253,264,265].

Technique		Output	Destructive	QC Method	General Comments
**Conformational assessments**	DSC	Thermal parameter (Tm, ΔG unfolding)	Yes	No	The best option for checking temperature-dependent parameters; not high throughput.
	CD	Secondary and tertiary statures	No	No	Sensitive to the polarity of the solution.
	Fluorescence spectroscopy	Tertiary statures/ general view on the protein structure	Yes	No	Sensitive to small structural changes of proteins and peptides; the need for relatively small amounts of material; relatively high speed; takes a general view of structural changes and cannot go into details.
	DSF	Tm, aggregation onset	Yes	No	During the thermal unfolding of a protein, a dye (commonly Sypro Orange) is added to the sample, which, by connecting to the unfolded parts, leads to an increase in its fluorescence emission.
	UV	Tertiary structure	No	No	Makes a general view of structural changes.
	Raman	Secondary structure/ chemical characterization	Yes	No	It has a high overlap with FTIR, but on the contrary, a wide range of solvents can be used in Raman analysis to examine samples.
	Infrared	Secondary structure/chemical integrity	Yes	No	By examining the amine-I and II regions of proteins and their deconvolution, it is possible to reach the percentage of the secondary structure in the protein (in solution and solid states) either in the form of fold and/or aggregation/fibrils. ATR facility requires very small amounts of substance without the need to prepare KBR tablets.
**Oligomerization studies**	AUC	Molecular weight/shape	No	Yes	Determining the size of particle aggregation.
	DLS	Hydrodynamic size		No	High range of particle detection (between 1nm to 5 µM), reliable within a certain range of polydispersity
	FAPS	Particles/Serum interactions		No	Extracting fluorescence data from the aggregated samples. One of its limitations is the use of dyes to identify aggregates, which may affect the protein structure.
	AF4	Hydrodynamic size	Yes	Yes	AF4 technique separates particles based on their diffusion coefficients.
	NTA	Hydrodynamic size	No		It can measure the size of particles, imaging and quantifying them.
	RMM	Concentration/size/mass	Yes	No	In a microfluidic way, it can calculate particle size.
	SEC	Hydrodynamic size	Yes	Yes	Determine the molecular weight, aggregation rate, and interactions between proteins.
	SDS-PAGE (all types)	Molecular weight/ interactions between proteins	Yes	Yes	Covalent interactions between proteins and also protein digests; in reducing and non-reducing types, it can observe disulfide bonds.
	Optical microscopy	Size/morphology	No	No	Detecting large particles (larger than 1 µm).
	Native MS	Fragments/aggregates	Yes	No	With principles similar to MS, it investigates non-covalent interactions and post-translational changes in proteins.
	Light obscuration	Concentration/size	No	Yes	This technique, which is also known as Single Particle Optical Sensing (SPOS), is not very sensitive to small sizes (detecting sizes more than 1 µm).
	Fluorescence microscopy	Particles/amorphous and morphous aggregates	Yes	No	A fluorophore molecule is needed that can provide the output signal. Some fluorophore dyes change fluorescence intensity by interacting with proteins and being buried in their structure, which can be a pattern of protein folding and even aggregation.
	Flow imaging	Concentration/size/morphology	No	Yes	The basis is similar to optical microscopy, except that it can provide data qualitatively.
	CE-SDS	Molecular weight	Yes	Yes	Advantages such as quicker analysis, the facility for quantification, full automation, the need for low sample, and better resolution
	Electrical zone sensing	Concentration/size	Yes	No	By applying electric force and migration of the two-pole magnifier, it can achieve the size of the particles.
	Turbidity	Optical density > 360 nm	No	Yes	It is a low-cost, rough method that can be used to detect large particles. Its high speed and simplicity are its positive points.
**Chemical changes**	RP-HPLC	Hydrophobicity	Yes	Yes	Sensitive to slight changes in surface hydrophobicity of proteins, requiring small amounts of samples.
	cIEF	Charge	Yes	Yes	Similar to IEF, it can separate proteins based on their PI. Compared to its traditional sample, i.e., IEF, it requires much less material, and due to its capillary nature, a higher voltage can be applied to the sample, which leads to a reduction in the test time. This test is successful in the case of samples higher than 150,000 Daltons that dissolve well in aqueous solutions.
	IEX chromatography	Charge	Yes	Yes	Separating proteins by considering their charge. High speed and acceptable accuracy.
	MS	By the difference in weight molecular changes	Yes	No	Differentiation of diverse components with considering mass-to-charge ratio (m/z).
	LC-MS	By the difference in weight molecular changes	Yes	No	In cases where a protein complex is present, initial separation by HPLC enables MS to obtain more details of the sample by removing noise.
	Zeta potential	Charge	No	Yes	Measuring the particle charges; the types of solvents and even the percentage of ions in water strongly affect the data.

Abbreviations: can: acetonitrile, AUC: analytical ultracentrifugation, DSC: differential calorimetry, Tm: melting temperature, CD: circular dichroism, DSF: differential scanning fluorimetry, UV: ultraviolet, DLS: dynamic light scattering, FAPS: fluorescence-activated particle sorter, AF4: asymmetrical flow-field flow fractionation, NTA: nanoparticle tracking analysis, RMM: resonant mass measurement, SEC: size-exclusion chromatography, SDS-PAGE: sodium dodecyl sulfate-polyacrylamide gel electrophoresis, MS: mass spectrometry, CE-SDS: capillary electrophoresis sodium dodecyl sulfate, RP-HPLC: reversed-phase high-performance liquid chromatography, cIEF: capillary isoelectric focusing, IEX: ion exchange chromatography, and LC: liquid chromatography.

## Data Availability

Not applicable.

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
