# Peer review of "Instability Challenges and Stabilization Strategies of Pharmaceutical Proteins"

_pharmaceutics, 2022, doi:10.3390/pharmaceutics14112533_

Round 1
Reviewer 1 Report
Excellent review and well written. The manuscript is ready for publication.
Author Response
We do appreciate your positive feedback and thank you for your kind words.
Reviewer 2 Report
Akbarian and Chen have comprehensively reviewed the factors affecting the stability of proteins that can be used in drug discovery. These reviews are organized around the overall topic so that the reader can quickly identify relevant areas. I believe that this manuscript will make a scientific contribution to readers in the relevant field. Before I support the publication of this paper, the following contents need to be revised by the author.
1. Typos
-line 12: 'side-drected mutagenesis' should be 'site-directed mutagenesis'
-line 26: ' variety of proteins and peptide' This sentence is not clear. Author should added some example.
-Line 178: ‘melting temperature of proteins (Tm). This temperature is’ melting temperature of proteins (Tm). This temperature is should be ‘melting temperature (Tm) of proteins. Tm is‘
-Lien 710: ‘HCL’ should be ‘HCl’’
-Lin4e 554, 558: ‘E. coli’ should be italic.
-Line 849: ‘as small angel X-ray crystallography a’ should be ‘as small angle X-ray scattering
-Line 923: ‘3D structure’ should be ‘folding’ based on author writing.
2. References to the sentences below should be clarified.
-Line 111-114: ‘Monoclonal antibodies (mAbs) accounted for almost half (48%) of the therapeutics protein sales in recent years. Since all the antibodies have a binding role to the antigen, the most important components in this category are maintaining the structure of complementarity-determining regions (CDRs) which bind their specific antigens.
-Line 633-644: ‘The linker is a very essential part of protein conjugation, especially for ADCs. Linker chemistry can be tuned to impart adequate stability for protein drugs. As mentioned above, PEGylation is an example of using hydrophilic PEG linkers to improve drug efficacy. For ADCs, the choice of linkers needs to possess sufficient stability in plasma so that ADC molecules can circulate in the bloodstream and localize to the tumor site without premature cleavage. At the same time, the linker needs to possess the ability to be rapidly cleaved and to release free and toxic payload once the ADC is internalized into the target tumor cell. There are cleavable and non-cleavable linkers. The cleavable linkers are cleaved by chemical or biochemical changes in the environment such as acidic changes for hydrozone linkers and glutathione or high pH values for disulfide linkers. Some other cleavable linkers contain enzymatic digestion sites and are sensitive to the enzymes inside the vesicle such as valine-citrulline di-peptides for cathepsin B enzyme.’
-Line 647-649: ‘The instability of the hydrazine linker and premature ozogamicin payload release was 647 hypothesized to be a primary reason for the poor efficacy of Mylotarg, which was 648 withdrawn from the market in 2010.’
-Line 685-702: ‘This process occurs unsurprisingly in nature. In the living system, various enzymes help a protein or peptide to react with a chain of saturated or unsaturated fatty acids in the form of N- and/or O-links. The binding site of lipids can be the terminal amine and carboxyl, and even in the middle part of the primary protein/peptide structure, which occurs through residues such as serine, threonine, glycine, lysine, and cysteine. It is interesting to note that this addition of lipid can be irreversible (when it is through the connection of the N-terminal of glycine/cysteine) or reversible (when a thioester bond occurs between the fatty acid and thiol group of cysteine). Regarding the medicinal aspects of lipidopeptides, some of these peptides are naturally extracted from microorganism sources such as fungi and bacteria. The use cases of this type of peptides have been proven in anti-cancer, anti-fungal and anti-bacterial fields. However, from another line of view, in silico addition of lipids to peptides is also of special interest. In addition to the fact that the presence of peptides can lead to the appearance of some new features in the peptide, it is reported that the type of spacer between the lipid and the peptide, the type of lipid and its chemical nature (saturated and unsaturated) are also effective in the activity of the lipidopeptide. The table below (Table 3) mentions some of the peptides that have been joined with lipids as encouraging options for treatment.’
-Line 762-770: ‘Another solution that has been used to preserve the integrity of medicinal proteins and peptides is the use of excipients during storage and even through the use of such drugs. According to the types of these agents, their mechanisms of action are also diverse. In the following table (Table 6), the types of these excipients are presented along with their mechanism to protect proteins. Some of these excipients should be used in a certain amount window with specific considerations such as salts that can lead to the aggregation of proteins after a critical concentration. Or as another important example, in the category of sugars, usually non-reducing sugars are used because reducing sugars via Millard’s reactions may cause destructive glycosylation of proteins.’
-Line 818-832: ‘Advances in analytical techniques have given researchers and biopharmaceutical companies new and increasingly sensitive methods for characterizing their products. As pharmaceutical proteins are very complex molecules, it is usually necessary to examine a particular aspect of stability by two or more techniques using an appropriate experimental design. The type of information obtained in such studies is highly dependent on the method used. Although the first sign of the validity of the protein may be its transparent state and its solubility, which can be easily seen even with the naked eye, most CQAs require more sensitive monitoring at the molecular level which needs to be examined by analytical instruments. Using LC-MS, subunit and intact protein masses can be directly measured to confirm product identity and quantify different sub-units, as well as to monitor conjugated species and the drug-to-antibody ratio of ADCs at the same time. Moreover, a bottom-up approach relying on the enzymatic digestion of the protein into peptides before LCMS is used to identify chemical modifications such as oxidation and deamidation on a specific residue site as well as to estimate their modification levels. Since’
-Line 882-890: ‘instability. Although these techniques require a lot of time to analyze the data, it gives a good experimental view of molecular dynamics. To summarize all the contents in this part, the following table (Table 7) tries to categorize the various techniques and methods of analyzing the structure of pharmaceutical proteins coherently. Many of these applied techniques have international standards from authorities such as the World Health Organization (WHO), Food and Drug Administration (FDA) and National Institutes of Health (NIH) to determine the accuracy of size. Also, these techniques must be evaluated and validated from the aspects of linearity, specificity, limit of detection, precision sensitivity recovery rate and accuracy.’
3. In the case of Figures 2, 3, 4 and 5, if the author refers to something that has been published elsewhere, the relevant reference should be added.
4. In the Tables, I don't think it's appropriate for the author to write long sentences. It should be abbreviated as keyword-centric words. And the author should draw a line in the horizontal direction to make it easier to see the items in the table.
5. The notation format of References is not uniform. Authors should make modifications to fit the format of the journal.
Author Response
Akbarian and Chen have comprehensively reviewed the factors affecting the stability of proteins that can be used in drug discovery. These reviews are organized around the overall topic so that the reader can quickly identify relevant areas. I believe that this manuscript will make a scientific contribution to readers in the relevant field. Before I support the publication of this paper, the following contents need to be revised by the author.
- Thank you ver much for your time and efforts. All your comments helped us to improve the quality of the study.
- Typos
-line 12: 'side-drected mutagenesis' should be 'site-directed mutagenesis'
- The correction was made (L12).
-line 26: ' variety of proteins and peptide' This sentence is not clear. Author should added some example.
- Some examples were added to the text (L26-27).
-Line 178: ‘melting temperature of proteins (Tm). This temperature is’ melting temperature of proteins (Tm). This temperature is should be ‘melting temperature (Tm) of proteins. Tm is‘
- The correction was made (L180).
-Lien 710: ‘HCL’ should be ‘HCl’’
- The correction was made (L735).
-Lin4e 554, 558: ‘E. coli’ should be italic.
- The correction was made (L585 and L589).
-Line 849: ‘as small angel X-ray crystallography a’ should be ‘as small angle X-ray scattering
- Thank you for pointing this out. The correction was made (L900).
-Line 923: ‘3D structure’ should be ‘folding’ based on author writing.
- The correction was made (L974).
- References to the sentences below should be clarified.
-Line 111-114: ‘Monoclonal antibodies (mAbs) accounted for almost half (48%) of the therapeutics protein sales in recent years. Since all the antibodies have a binding role to the antigen, the most important components in this category are maintaining the structure of complementarity-determining regions (CDRs) which bind their specific antigens.
- The relevant reference was added (Reference 24).
-Line 633-644: ‘The linker is a very essential part of protein conjugation, especially for ADCs. Linker chemistry can be tuned to impart adequate stability for protein drugs. As mentioned above, PEGylation is an example of using hydrophilic PEG linkers to improve drug efficacy. For ADCs, the choice of linkers needs to possess sufficient stability in plasma so that ADC molecules can circulate in the bloodstream and localize to the tumor site without premature cleavage. At the same time, the linker needs to possess the ability to be rapidly cleaved and to release free and toxic payload once the ADC is internalized into the target tumor cell. There are cleavable and non-cleavable linkers. The cleavable linkers are cleaved by chemical or biochemical changes in the environment such as acidic changes for hydrozone linkers and glutathione or high pH values for disulfide linkers. Some other cleavable linkers contain enzymatic digestion sites and are sensitive to the enzymes inside the vesicle such as valine-citrulline di-peptides for cathepsin B enzyme.’
- The relevant references were added (References 151, 153).
-Line 647-649: ‘The instability of the hydrazine linker and premature ozogamicin payload release was hypothesized to be a primary reason for the poor efficacy of Mylotarg, which was withdrawn from the market in 2010.’
- The relevant references were added (References 154, 155).
-Line 685-702: ‘This process occurs unsurprisingly in nature. In the living system, various enzymes help a protein or peptide to react with a chain of saturated or unsaturated fatty acids in the form of N- and/or O-links. The binding site of lipids can be the terminal amine and carboxyl, and even in the middle part of the primary protein/peptide structure, which occurs through residues such as serine, threonine, glycine, lysine, and cysteine. It is interesting to note that this addition of lipid can be irreversible (when it is through the connection of the N-terminal of glycine/cysteine) or reversible (when a thioester bond occurs between the fatty acid and thiol group of cysteine). Regarding the medicinal aspects of lipidopeptides, some of these peptides are naturally extracted from microorganism sources such as fungi and bacteria. The use cases of this type of peptides have been proven in anti-cancer, anti-fungal and anti-bacterial fields. However, from another line of view, in silico addition of lipids to peptides is also of special interest. In addition to the fact that the presence of peptides can lead to the appearance of some new features in the peptide, it is reported that the type of spacer between the lipid and the peptide, the type of lipid and its chemical nature (saturated and unsaturated) are also effective in the activity of the lipidopeptide. The table below (Table 3) mentions some of the peptides that have been joined with lipids as encouraging options for treatment.’
- The relevant references were added (References 164-170).
-Line 762-770: ‘Another solution that has been used to preserve the integrity of medicinal proteins and peptides is the use of excipients during storage and even through the use of such drugs. According to the types of these agents, their mechanisms of action are also diverse. In the following table (Table 6), the types of these excipients are presented along with their mechanism to protect proteins. Some of these excipients should be used in a certain amount window with specific considerations such as salts that can lead to the aggregation of proteins after a critical concentration. Or as another important example, in the category of sugars, usually non-reducing sugars are used because reducing sugars via Millard’s reactions may cause destructive glycosylation of proteins.’
- The relevant references were added (References 180-182).
-Line 818-832: ‘Advances in analytical techniques have given researchers and biopharmaceutical companies new and increasingly sensitive methods for characterizing their products. As pharmaceutical proteins are very complex molecules, it is usually necessary to examine a particular aspect of stability by two or more techniques using an appropriate experimental design. The type of information obtained in such studies is highly dependent on the method used. Although the first sign of the validity of the protein may be its transparent state and its solubility, which can be easily seen even with the naked eye, most CQAs require more sensitive monitoring at the molecular level which needs to be examined by analytical instruments. Using LC-MS, subunit and intact protein masses can be directly measured to confirm product identity and quantify different sub-units, as well as to monitor conjugated species and the drug-to-antibody ratio of ADCs at the same time. Moreover, a bottom-up approach relying on the enzymatic digestion of the protein into peptides before LCMS is used to identify chemical modifications such as oxidation and deamidation on a specific residue site as well as to estimate their modification levels. Since’
- The corresponding references were added (References 243-246).
-Line 882-890: ‘instability. Although these techniques require a lot of time to analyze the data, it gives a good experimental view of molecular dynamics. To summarize all the contents in this part, the following table (Table 7) tries to categorize the various techniques and methods of analyzing the structure of pharmaceutical proteins coherently. Many of these applied techniques have international standards from authorities such as the World Health Organization (WHO), Food and Drug Administration (FDA) and National Institutes of Health (NIH) to determine the accuracy of size. Also, these techniques must be evaluated and validated from the aspects of linearity, specificity, limit of detection, precision sensitivity recovery rate and accuracy.’
- The corresponding references were added (References 10, 263).
- In the case of Figures 2, 3, 4 and 5, if the author refers to something that has been published elsewhere, the relevant reference should be added.
- The corresponding references were added (References 21, 38 for Figure 1. Ref. 54 for Figure 2. Ref. 63 for Figure 3. Ref. 79, 80 for Figure 4. ).
- In the Tables, I don't think it's appropriate for the author to write long sentences. It should be abbreviated as keyword-centric words. And the author should draw a line in the horizontal direction to make it easier to see the items in the table.
- Thank you for your suggestion. All tables were amended accordingly.
- The notation format of References is not uniform. Authors should make modifications to fit the format of the journal.
- Thank you for pointing this out. We checked all references to correct any disorder.
Reviewer 3 Report
The review by Akbarian and Chen discusses the challenges associated with protein/peptide stability for pharmaceutical products. The review is extensive and I found it very informative, thanks to the authors.
My main remarks are mostly on the organisation of the content.
1. The presentation of the issues is pretty didactic, thanks again to the authors, but the manuscript may lack an overall scheme to present the issues discussed and position the strategies to overcome them. As presently, it is difficult to identify at a glance what is discussed in this review.
2. The flow is sometimes rather dense. There are four figure for section 3, 2 in section 4, and none in section 5. Better balancing of the illustrations could help.
3. There must be some differences between issues encountered for large proteins and those encountered for instance for small peptides. These differences are not clearly stated and discussed, nor are variations in strategy to overcome specific problems, if any.
4. From the title, I was expecting some analysis on how the different strategies can or cannot be employed depending on each specific case. Could the authors elaborate a bit (more) on this ? I am with the impression that some general decision flowchart could be sketched, to the benefit of the reader.
Author Response
The review by Akbarian and Chen discusses the challenges associated with protein/peptide stability for pharmaceutical products. The review is extensive and I found it very informative, thanks to the authors.
- We eternally appreciate your feedback.
My main remarks are mostly on the organisation of the content.
- The presentation of the issues is pretty didactic, thanks again to the authors, but the manuscript may lack an overall scheme to present the issues discussed and position the strategies to overcome them. As presently, it is difficult to identify at a glance what is discussed in this review.
- Thank you for drawing attention to this. We have revised the abstract and Introduction to highlight the review focuses (L14-18 and L33-80)
- The flow is sometimes rather dense. There are four figure for section 3, 2 in section 4, and none in section 5. Better balancing of the illustrations could help.
- We are very grateful for your comment. A new figure that expresses various analytical methods used to detect and characterize protein instability has been added to Figure 9.
- There must be some differences between issues encountered for large proteins and those encountered for instance for small peptides. These differences are not clearly stated and discussed, nor are variations in strategy to overcome specific problems, if any.
- With many thanks for mentioning this point, there are differences in the susceptibility of proteins according to their size to environmental stresses, which are mentioned in L460-490. Nonetheless, for some stresses that depend on the amino acid sequence, for example, the possibility of the presence of this sequence (independent of the protein size) leads to their vulnerability.
- From the title, I was expecting some analysis on how the different strategies can or cannot be employed depending on each specific case. Could the authors elaborate a bit (more) on this ? I am with the impression that some general decision flowchart could be sketched, to the benefit of the reader.
- Your comment brought to our mind to draw a general flowchart at the end of Section 4 (Figure 8). In this figure, in the beginning, we have divided the proteins into peptides, proteins, and ADCs, then, we have proposed a solution according to the stresses that usually threaten them (L800-803).
Round 2
Reviewer 3 Report
Thank you for considering the remark in a straight manner.